



# Development of daily PM₁₀ and PM₂.₅ prediction system using a deep long short-term memory neural network model

Hyun S. Kim[1,†], Inyoung Park[2,†], Chul H. Song[1], Kyunghwa Lee[1], Jae W. Yun[2], Hong K. Kim[2], Moongu Jeon[2], Jiwon Lee[2]

5 [1]School of Earth Sciences and Environmental Engineering, Gwangju Institute of Science and Technology (GIST), Gwangju 61005, Korea
[2]School of Electrical Engineering and Computer Science, Gwangju Institute of Science and Technology (GIST), Gwangju 61005, Korea
[†]Both authors are equally contributed to this work

10 *Correspondence to*: Chul Han Song (chsong@gist.ac.kr)

**Abstract.** A deep recurrent neural network system based on a long short-term memory (LSTM) model was developed for daily PM₁₀ and PM₂.₅ predictions in South Korea. The structural and learnable parameters of the newly developed system were optimized from iterative model trainings. Independent variables were obtained from ground-based observations over 2.3 years. The performance of the particulate matter (PM) prediction LSTM was then evaluated by comparisons with ground 15 PM observations and with the PM concentrations predicted from two sets of 3-D chemistry-transport model (CTM) simulations (with and without data assimilation for initial conditions). The comparisons showed, in general, better performance with the LSTM than with the 3-D CTM simulations. For example, in terms of IOAs (index of agreements), the PM prediction IOAs were enhanced from 0.36-0.78 with the 3-D CTM simulations to 0.62-0.79 with the LSTM-based model. The deep LSTM-based PM prediction system developed at observation sites is expected to be further integrated with 3-D 20 CTM-based prediction systems in the future. In addition to this, further possible applications of the deep LSTM-based system are discussed, together with some limitations of the current system.

## 1 Introduction

Over the past several decades, South Korea has made continuous economic growth; however, in accordance with this rapid economic development, emissions of air pollutants from various sources such as industrial, transportation, and power 25 generation sectors have increased, and air quality has thus deteriorated (Wang et al., 2014). Among the atmospheric pollutants, particulate matter (PM) plays an important role in human health and climate change (Davidson et al., 2005; Forster et al., 2007). Several epidemiological studies have reported clear statistical relationships between aerosol concentrations and human mortality and morbidity (Dockery et al., 1992; Hope III and Dockery, 2006). To minimize the public damage caused by air pollution and to alert Korean citizen about high PM events, the National Institute of



Environmental Research (NIER) of South Korea has carried out daily air quality (or chemical weather) forecasting using multiple 3-D chemistry-transport models (CTMs) since 2014.

However, the accuracy of the 3-D CTM simulations has been reported to be low. Researchers believe that this low accuracy originates from uncertain sources of emission inventory, meteorological fields, initial and boundary conditions, and CTMs

themselves (Seaman, 2000; Berge et al., 2001; Liu et al., 2001; Holloway et al., 2008; Tang et al., 2009; Han et al., 2011). Many efforts have been made to enhance the accuracy of the 3-D CTM-based forecasting system. As a part of the efforts, the Korean government decided to develop its own air quality forecasting system mainly based on a new CTM in 2017. This project entailed establishing better bottom-up and top-down emissions, developing improved meteorological fields over East Asia, developing a data assimilation system using satellite-retrieved and ground-based observations, and incorporating new

atmospheric chemical/physical processes into the new Korean CTM. Despite all the ongoing efforts, the traditional chemical weather forecasts based on the CTM are still poor at conducting accurate air quality forecasts over South Korea.

In contrast, statistical models based on artificial neural networks (ANNs) have also been applied to air quality predictions. Because these approaches are based on a statistical method instead of sophisticated mathematical model-based computations (i.e., without considerations of advection/convection, photochemistry, or emissions), they are more cost-effective than 3-D

CTM simulations. In previous studies, simple ANN models were applied to air quality predictions. The time-series concentrations of ambient pollutants have been predicted by, for example, supported vector machine (SVM) and radial basis function (RBF) neural network models (Lu and Wang, 2005). Furthermore, ambient levels of ozone were predicted by a simple feed-forward neural network (FFNN) models (Yi and Rybutok, 1996; Abdul-Wahab and Al-Alawi, 2002). However, such simple models have the limitation of neglecting relationships among data at the different time steps. Recently, more

complex ANN models have been developed with recurrent neural networks (RNNs). Although RNNs have typically been used for natural language recognition, they have the special advantage of remembering the experiences of past events because they maintain the activated vectors at each time step (Cho et al., 2014). Because of this advantage, RNNs also make accurate time-series predictions (Che et al., 2018). Several investigators used a shallow (single hidden layer) RNN model to predict the peak mixing ratios of ambient pollutants such as $NO_2$, $SO_2$, $O_3$, CO, and $PM_{10}$ (Brunelli et al., 2007), and others

used a deep RNN model to predict ambient levels of $PM_{2.5}$ (Ong et al., 2016). However, RNN models have generally shown serious exploding and/or vanishing gradient problems (Bengio et al., 1994; Hochreiter, 1998). To resolve these problems, researchers developed the long short-term memory (LSTM) cell (Hochreiter and Schmidhuber, 1997). Unlike traditional RNNs, LSTM is known to be free from exploding or vanishing gradient problems, and it is better suited for long time-series predictions than are traditional RNNs. Recently, researchers used a deep LSTM neural network to conduct a number of air

quality studies (Li et al., 2017; Freeman et al., 2018).

Although ANN-based predictions are not mathematics-based, deep learning has demonstrated strong potential in the areas of weather and air quality forecasts; for example, the Weather Channel in the United States uses IBM Watson for its operational weather predictions (Mourdoukoutas, 2015). Another example is bias corrections based on several machine-learning techniques. Authors of one study reported that the biases (or errors) between the operational CTM-based air quality





predictions and observations can be reduced by utilizing machine learning algorithms (Reid et al., 2015). There must be many creative ways to improve the accuracy of air quality forecasts by combining 3D CTM-based predictions with artificial intelligence (AI)-based techniques. These combined approaches have now begun, and this manuscript intends to present one of these efforts in the area of air quality predictions.

For this study, we developed a deep LSTM model to more accurately predict ambient PM concentrations. We evaluated the model performance by comparing the CTM-predicted and observed $PM_{10}$ and $PM_{2.5}$ with the LSTM-predicted $PM_{10}$ and $PM_{2.5}$. The details of the system development and prediction procedures are presented in Sects. 2 and 3, and limitations of the model are discussed in Sect. 4.

## 2 Model development

Fig. 1 shows the schematic procedures for the deep LSTM model-based PM predictions. There were two main processes in developing this prediction system: (i) data preprocessing and (ii) structure design and optimization of the deep neural network. It is essential to prepare time-series sequential data sets for both model training and predictions. In this study, we collected ambient pollutant concentrations and meteorological data from ground-based observations. To construct the system, we first screened several AI-based methods including LSTM, such as SVM, relevance vector machine (RVM), and a

technique from convolutional neural network (CNN). Based on the results from the screening, we chose a multi-layered deep LSTM neural network and conducted iterative model training to optimize the model weights and biases. We present the details on developing the system in the following sections.

### 2.1 Data preprocessing

We collected the observation data from both the NIER AIR KOREA measurement network and the Korea Metrological

Administration (KMA) automatic weather station (AWS) network to prepare the input variables. Fig. 2 presents the locations of the AIR KOREA and KMA AWS observation sites throughout South Korea; the networks consist of 323 and 494 ground-based monitoring stations, respectively. They provide hourly mixing ratios of the ambient pollutants such as $SO_2$, CO, $NO_2$, $O_3$, $PM_{10}$, and $PM_{2.5}$ and the metrological parameters such as temperature, wind direction, wind speed, hourly precipitation, and relative humidity. Both $PM_{10}$ and $PM_{2.5}$ are measured by β-ray absorption and gravimetric method, respectively (Shin et

al., 2011). The ambient mixing ratios of $SO_2$, CO, $NO_2$, and $O_3$ are measured by pulse ultraviolet fluorescence, non-dispersive infrared, chemiluminescence, and ultraviolet methods, respectively.

Among the observation sites, we chose seven monitoring sites located in the major cities in South Korea (two sites in Seoul, Daejeon, Gwangju, Daegu, Ulsan, and Busan) for $PM_{10}$ and $PM_{2.5}$ predictions (refer to Fig. 2 (a)-(f) regarding the locations). There were two main criteria in our selecting the seven sites: (i) the distances between air quality and meteorological

monitoring stations should be the shortest (i.e., collocation), and (ii) the number of missing observation data should be minimal. Because there are sometimes too many missing values in the monitoring data prior to 2013, we used observations





from January 2014 to April 2016 for training. After the model training, actual predictions of $PM_{10}$ and $PM_{2.5}$ were conducted for the period of the Korea-United States Air Quality (KORUS-AQ) campaign (from May 1 to June 11, 2016). The KORUS-AQ campaign period is now an official model testing window in South Korea.

High quality of input data is critical for LSTM-based time-series predictions. In the current study, the missing values in
ground-based air quality monitoring data were produced by using the pre-trained deep LSTM model. The schematic diagram of missing value generation is presented in Fig. S1. As shown in Fig. S1, when the missing data were detected, the corresponding values were generated from a pre-trained model. For example, the accuracy of the missing values generated for Seoul-1 site is summarized in Fig. S2. It is shown from Fig. S2 that the pollutant concentrations generated by the pre-trained model correlated well with the observed concentrations. The correlation coefficients for the model training and
validation ranged from 0.60 to 0.91 and from 0.52 to 0.93, respectively. The accuracy of the generated missing values from the seven selected monitoring stations is summarized in Table S1. For the meteorological parameters, we determined the missing variables by interpolating the observed data; in the meteorological data, fewer than 0.01% of values were missing.

In particular, information on various pollutants is important in the LSTM-based predictions of $PM_{10}$ and $PM_{2.5}$. Because $H_2SO_4$ and $HNO_3$ are main precursors of inorganic sulfate ($SO_4^{2-}$) and nitrate ($NO_3^-$), respectively, correct information on the
levels of their precursors ($SO_2$ and $NO_2$) is important. Although CO is not directly related to producing particulate matter, we included the mixing ratios in the input data because these are somehow related to the mixing ratios of ozone and hydroxyl radicals (OH).

Meteorological conditions also play an important role in particulate matter concentrations. Both wind direction and speed can represent the origin of air pollutants and intensity of atmospheric turbulences, and precipitation directly affects $PM_{10}$ and
$PM_{2.5}$ by wet scavenging. In addition, there is a relationship between relative humidity and the levels of hydroxyl radicals because $H_2O$ is a main precursor of OH. Water vapor can also influence the amounts of particulate water and nucleation rates in the atmosphere. Therefore, all meteorological parameters measured in the AWS monitoring data set can possibly affect PM concentrations, and thus we used them in the LSTM-based prediction system.

Before feeding the input variables into the LSTM system, one important step is data normalization. All the input parameters
were rescaled between 0 and 1:

$$x_{normal,i} = \frac{x_i - x_{\min,i}}{x_{\max,i} - x_{\min,i}} \tag{1}$$

where $x_{normal,i}$ is the normalized values of species $i$; $x_i$ is the observed value; $x_{\max,i}$ and $x_{\min,i}$ are the maximum and minimum values of species $i$, respectively. We reshaped the normalized variables as a three-dimensional vector matrix to feed them into the hidden LSTM layers. Because we designed the system for daily forecasting of $PM_{10}$ and $PM_{2.5}$, the step
size was 24. In addition, we excluded the observation data during the dust event periods in the model training; because these





episodes are infrequent, including data on them could have interfered with establishing an accurate PM prediction system (i.e., they can be noisy signals).

## 2.2 System construction

The developed system has two schemes: $PM_{10}$ prediction and $PM_{2.5}$ prediction; the prediction model is designed to have

three to five hidden LSTM layers; one layer consists of one hundred hidden nodes, and the layers capture sequential temporal information. The last LSTM hidden layer is connected to the output layer, which performs feature mapping between the output vectors from deep hidden layers and the actual $PM_{10}$ and/or $PM_{2.5}$.

In order to learn complicated and nonlinear mappings between the layers, activation functions are applied to get the output of a layer, which is then fed into the next layer as an input. There are several activation functions that can be used in neural

networks; among them, a sigmoid function has been typically used because it has characteristics of being bounded and being differentiable; however, this function has a vanishing gradient problem due to continuous multiplication of gradients. In this study, therefore, we used the rectified linear unit (ReLU) to activate output layer (Nair and Hinton, 2010). The ReLU is expressed by:

$$f(x) = \begin{cases} 0 & when\ x < 0 \\ x & when\ x \geq 0 \end{cases} \tag{2}$$

As shown in Eq. (2), the ReLU ranges from 0 to $\infty$. Because the derivative of the ReLU is 0 or 1, the vanishing gradient does not occur during the back propagations. In Supplementary Material (SM), we give a detailed description of the LSTM architecture used in this study.

## 2.3 Model training

The model training is a process for optimizing the structural and learnable parameters of the deep LSTM system; we

determined the $PM_{10}$ and $PM_{2.5}$ prediction system's structure from the iterative trainings, and the structure was described in Sect. 2.2. In addition to the activation functions described in Sect. 2.2, there are two more main components in the deep neural network training: (i) cost function and (ii) optimization algorithm. The cost function usually measures how well the neural network works with respect to given training samples and corresponding predicted outputs. In other words, it is used for evaluating the accuracy of predicted values. When the prediction accuracy is poor, the cost is high, whereas as the

model's predictions are more accurate, the cost decreases.

There are several cost functions commonly used in deep learning, and the cost function can be classified by its application purpose. In this study, the purpose of the cost function was to minimize the regression cost, and we thus used mean squared error (MSE) as a cost function, expressed as:

$$J_{MSE}(\theta) = \frac{1}{N} \sum_{i=1}^{N} (y_i - h_\theta(x_i))^2 \tag{3}$$



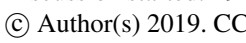

where $x_i$ is the input vector for $i^{th}$ training; $y_i$ is the target value (or true value) for the $i^{th}$ training; and $h_\theta(x_i)$ is the predicted vector corresponding to $y_i$ for a given deep neural network model $\theta$. It should be noted here that $\theta$ means the LSTM network with different parameters (see Eqs. (S1)-(S6) in SM). In Eq. (3), $J_{MSE}(\theta)$ represents the MSE between the target vector, $y_i$, and its predicted vector, $h_\theta(x_i)$, when the number of training vectors is $N$.

The role of an optimization algorithm is to find an efficient and stable pathway for minimizing the gradient descent of a cost function. In this study, we utilized adaptive moment estimation (ADAM) to train the neural networks (Kingma and Ba, 2015). The ADAM is one of the extended algorithms for stochastic gradient descent, and its detailed explanation is also given in SM.

In order to train the LSTM system for the PM$_{10}$ and PM$_{2.5}$ predictions, the observations from January 2014 to April 2016
(2.3-year data) were used, as mentioned previously. Because Asian dust was regarded as noisy atmospheric signals, we removed the observations during dust events in the course of the model training. We divided the training data set into two groups with ratios of 0.85 to 0.15 for the model training and validation, respectively (Guyon, 1997).

First, we measured the accuracy of the trained and optimized LSTM system using two statistical parameters, MSE and root mean squared error (RMSE); these values are summarized in Table 1. The model training can be verified by comparing the
training and validation cost (i.e., the outcome of MSE cost); for properly trained models, the training cost should be smaller than the validation cost. If the training cost is higher, it is called "over-fitting", which means that the weight and bias vectors of the model have been overturned. Based on the statistical analysis, we concluded that the model was well trained via general rules of the model training (refer to Table 1).

Second, the accuracy of the deep LSTM model training for PM$_{10}$ and PM$_{2.5}$ predictions is summarized in Figs. 3 and 4,
respectively; in the figures, the black and red dots denote the comparison results between the predicted and observed PM concentrations in the model training and validation, respectively. As shown in Fig. 3, there was reasonable agreement in the PM$_{10}$ predictions. The Pearson correlation coefficients (R) for PM$_{10}$ training and validation ranged from 0.61 to 0.81 and from 0.55 to 0.71, respectively. The training results for the PM$_{2.5}$ model also showed reasonable correlations ($0.59 \leq R$ for training $\leq 0.80$; $0.54 \leq R$ for validation $\leq 0.75$).

**2.4 3-D CTM simulations**

In order to assess the accuracy of the LSTM-based predictions in this study, we compared them with 3-D CTM-based predictions with and without data assimilation (DA). We employed the Community Multiscale Air Quality (CMAQ) model v5.1 for the 3-D CTM simulations. We acquired the metrological fields from Weather Research and Forecasting v3.8.1 model simulations. The domain of the CMAQ model simulations is presented in Fig. 5; the model domain covers Northeast
Asia with a horizontal resolution of 15×15 km$^2$ and with 27 sigma vertical levels. We used the KORUS v1.0 emission inventory for anthropogenic emissions; this inventory was made for the KORUS-AQ campaign based on three emission





inventories: (i) CREATE (Comprehensive Regional Emission inventory for Atmospheric Transport Experiment); (ii) MICS-Asia (Model Inter-Comparison Study for Asia); and (iii) SEAC4RS (Studies of Emissions and Atmospheric Composition, Clouds, and Climate Coupling by Regional Surveys) (Woo et al., 2017). We estimated biogenic emissions from MEGAN v2.1 (Model of Emissions of Gases and Aerosols from Nature) simulations (Guenther et al., 2006). We obtained biomass

burning emissions from FINN (Fire INventory from NCAR, http://bai.acom.ucar.edu/Data/fire/) (Wiedinmyer et al., 2011). We obtained lateral boundary conditions from the MOZART-4 model simulations (https://www.acom.ucar.edu/wrf-chem/mozart.shtml) (Emmons et al., 2010).

To prepare the initial conditions (ICs) for the CMAQ model simulations, we used optimal interpolation method with Kalman filter (OI with Kalman). The DA with the OI technique has been used in several previous studies (Carmichael et al., 2009;

Chung et al., 2010). The assimilation system is defined as follows:

$$\tau_m^{'} = \tau_m + K\left(\tau_o - H\tau_m\right) \tag{4}$$

$$K = BH^{T}\left(HBH^{T} - O\right)^{-1} \tag{5}$$

where $\tau_m^{'}$ represents the assimilated product; $\tau_o$ and $\tau_m$ denote the observed and modeled values, respectively; $H$ represents the observation and/or forward operator; $K$ represents the Kalman gain matrix; and $B$ and $O$ are the covariance

of modeled and observed field, respectively. $B$ and $O$ can be defined with several free parameters.

$$B(d_x, d_z) = \left[(f_m \tau_m)^2 + (\varepsilon_m)^2\right]\exp\left[-\frac{d_x^2}{2l_{mx}^2}\right]\exp\left[-\frac{d_z^2}{2l_{mz}^2}\right] \tag{6}$$

$$O = \left[(f_o \tau_o)^2 + (\varepsilon_o)^2\right]I \tag{7}$$

where $f_m$, $\varepsilon_m$, $d_x$, $d_z$, $l_{mx}$, and $l_{mz}$ represent the fractional error coefficient, minimum error coefficient, horizontal resolution, vertical resolution, horizontal correlation length for errors in modeled values, and vertical correlation length for

errors in modeled values, respectively, and $f_o$, $\varepsilon_o$, and $I$ denote the fractional error coefficient, minimum error coefficient in the observed values, and unit matrix, also respectively. The six parameters in Eq. (6) are called the free parameters, which we used to calculate the observation and model error covariance matrix, respectively. In this study, we determined these parameters by finding the minimum of $\chi^2$:

$$\chi^2 = \sum \frac{(x_{obs} - x_{assim})^2}{x_{assim}} \tag{8}$$





here $x_{obs}$ and $x_{assim}$ represent the observed and data-assimilated values. More detailed explanations regarding the data assimilation can be found elsewhere (Collins et al., 2001; Yu et al., 2003; Park et al., 2011).

For the DA runs, we integrated the CMAQ model simulations with three observation data sets: (i) the Communication, Ocean, and Meteorological Satellite (COMS)/Geostationary Ocean Color Imager (GOCI) aerosol optical depth (AOD); (ii)
ground-based observations in China; and (iii) AIR KOREA observations in South Korea. The locations of the observation stations are presented in Fig. 5. Because the GOCI sensor is geostationary, it can provide hourly spectral images with spatial resolution of 500×500 m$^2$ from 00:00 to 07:00 UTC. Detailed procedures can also be found in previous publications (Park et al., 2014; Park et al., 2014).

## 3 Results and Discussion

PM$_{10}$ and PM$_{2.5}$ were predicted for the period of the KORUS-AQ campaign. To evaluate the performance of the LSTM system, we compared the LSTM-based predictions with the observations and two CMAQ-based predictions.

### 3.1 System evaluation

We evaluated the accuracy of the LSTM-based PM predictions by comparing them with the observed PM$_{10}$ and PM$_{2.5}$. We also compared PM$_{10}$ and PM$_{2.5}$ predicted from two sets of CMAQ model simulations with the PM$_{10}$ and PM$_{2.5}$ predicted from
the deep LSTM; these results are presented in Figs. 6 and 7. In Figs. 6 and 7, the black circles and blue-dashed lines represent the observed and LSTM-predicted PM$_{10}$ and PM$_{2.5}$, respectively. The green- and red-dashed lines denote CMAQ-predicted PM$_{10}$ and PM$_{2.5}$ with and without DA, respectively. The CMAQ model simulations with DA showed better agreement with the observations than did those without DA (see Figs. 6 and 7). The LSTM-predicted PM$_{10}$ also showed good agreement with the observed PM$_{10}$.

For further statistical evaluations, we introduced the following five statistical parameters: (i) IOA; (ii) RMSE; (iii) MB (Mean Bias); (iv) MNGE (Mean Normalized Gross Error); and (v) MNB (Mean Normalized Bias).

$$IOA = 1 - \frac{\sum_{1}^{N}\left(C_{i,Model} - C_{i,Obs}\right)^2}{\sum_{1}^{N}\left(\left|C_{i,Model} - \overline{C_{i,Obs}}\right| + \left|C_{i,Obs} - \overline{C_{i,Obs}}\right|\right)^2} \qquad (9)$$

$$RMSE = \sqrt{\frac{1}{N}\sum_{1}^{N}\left(C_{i,Model} - C_{i,Obs}\right)^2} \qquad (10)$$

$$MB = \frac{1}{N}\sum_{1}^{N}\left(C_{i,Model} - C_{i,Obs}\right) \qquad (11)$$



$$MNGE = \frac{1}{N}\sum_{1}^{N}\left(\frac{\left|C_{i,Model}-C_{i,Obs}\right|}{C_{i,Obs}}\right)\times 100 \tag{12}$$

$$MNB = \frac{1}{N}\sum_{1}^{N}\left(\frac{C_{i,Model}-C_{i,Obs}}{C_{i,Obs}}\right)\times 100 \tag{13}$$

Here, $C_{i,Model}$ and $C_{i,Obs}$ represent the modeled and observed concentrations of species $i$; $\overline{C_{i,Obs}}$ is the averaged $C_{i,Obs}$. The results from the statistical analysis are summarized in Table 2 and are also shown in Figs. 6 and 7.

For the daily $PM_{10}$ predictions, the LSTM-based predictions ($0.62 \leq IOA \leq 0.79$) were always more accurate than two CMAQ-based $PM_{10}$ predictions ($0.36 \leq IOA \leq 0.78$). RMSE and MB between the CMAQ-based and observed $PM_{10}$ ranged between 33.11 μg/m$^3$ and 51.40 μg/m$^3$ and between -40.35 μg/m$^3$ and -15.95 μg/m$^3$, respectively. These negative MBs indicate that the CMAQ model simulations underestimated $PM_{10}$. RMSE and MB between the LSTM-based predictions and observations ranged from 18.57 μg/m$^3$ to 24.23 μg/m$^3$ and from -3.20 μg/m$^3$ to 6.28 μg/m$^3$, respectively. The RMSEs for the

LSTM-based predictions are 1.90 times smaller than those for the CMAQ-based predictions. Among the seven sites chosen, the $PM_{10}$ predictions at the Daegu, Ulsan, and Busan sites showed the best agreement ($0.71 \leq IOA \leq 0.79$) and the lowest errors and biases ($16.46 \leq RMSE \leq 18.57$; $-1.00 \leq MB \leq 6.02$) compared with the two CMAQ-based $PM_{10}$ predictions ($0.45 \leq IOA \leq 0.65$; $26.23 \leq RMSE \leq 55.09$; $-37.69 \leq MB \leq -15.92$).

Fig. 7 presents the comparisons for $PM_{2.5}$. During the KORUS-AQ campaign, there were no ground $PM_{2.5}$ observations at the

15 Daejeon site between May 1 and June 11 because of instrument malfunction. The LSTM-predicted $PM_{2.5}$ again showed good agreement with the observations ($0.63 \leq IOA \leq 0.79$); however, the deep LSTM system was not always able to more accurately predict $PM_{2.5}$. As with $PM_{10}$, the LSTM $PM_{2.5}$ predictions at the Daegu, Ulsan, and Busan sites showed better performance ($0.78 \leq IOA \leq 0.79$) than the CMAQ-based predictions ($0.59 \leq IOA \leq 0.75$), but at the two Seoul sites (Seoul-1 and Seoul-2), the LSTM $PM_{2.5}$ predictions were inferior to those from the CMAQ model simulations with DA (green-dashed

lines in Fig. 7). This could have been because the AIR KOREA observation sites are densely located in and around Seoul Metropolitan Area (refer to Fig. 2). Therefore, data assimilation appears to more strongly influence the accuracy of the CMAQ predictions.

As shown in Figs. 6 and 7, there were nationwide high PM episodes from May 25 to 28, 2016; these high PM events were caused by long-distance transport of atmospheric pollutants from China due to westerlies, and the relatively high errors and

25 biases in the LSTM-based predictions occurred during these high PM events. Because the model's weights and biases were optimized based on previous memories, frequent high PM episodes can affect the accuracy of the predictions. The frequencies of the high $PM_{10}$ (daily average of $PM_{10} \geq 70$ μg/m$^3$) and high $PM_{2.5}$ (daily average of $PM_{2.5} \geq 40$ μg/m$^3$) episodes in the training data set are summarized in Fig. 8. The fractions of the high $PM_{2.5}$ episodes at the Seoul and Gwangju





sites were between 0.04 and 0.09, clearly smaller than those at Daegu, Ulsan, and Busan (0.12 ≤ high $PM_{2.5}$ episode ≤ 0.18). At Gwangju, the effectiveness of DA and frequency of high PM episodes were the lowest. As mentioned previously, the LSTM-based $PM_{10}$ and $PM_{2.5}$ prediction system was trained using the observation data for only 2.3 years because these were the only available data. The optimized weights and biases are governed by the variety of input features in the training. If

more $PM_{2.5}$ data are available in the future, the prediction accuracy of deep LSTM systems will improve, and in fact, continuous data accumulation with more recent PM data is now underway.

**3.2 Dependence on input parameters**

In deep learning, the relationships between input variables and predictions cannot be identified directly because of the high non-linearity in the hidden layers. In the present study, we indirectly investigated the influences of the input parameters on

the $PM_{10}$ and $PM_{2.5}$ predictions with and without considering each variable in the model operations. The influences on the input parameters are summarized in Figs. 9 and 10; in the figures, TA, WD, WS, RN, RNH, and RH represent temperature, wind direction, wind speed, daily cumulative precipitation, hourly precipitation, and relative humidity, respectively. $SO_2$, $O_3$, $NO_2$, CO, $PM_{10}$, and $PM_{2.5}$ are the concentrations of the respective air pollutants on the previous day. The positive and negative values in each figure represent the directionality of the influences on the $PM_{10}$ and $PM_{2.5}$ predictions; that is, for

instance, the variables with positive dependence indicate increasing influence on the predicted $PM_{10}$ and $PM_{2.5}$. The figures show that among the meteorological variables, temperature and wind direction generally had great influence on the $PM_{10}$ and $PM_{2.5}$ predictions; among the pollutant variables, previous day's $PM_{10}$ and $PM_{2.5}$ mainly affected the predictions for the next day. In particular, the dependencies of $PM_{10}$ and $PM_{2.5}$ ranged from 38.48 % to 60.12 % and from 28.80 % to 83.38 %, respectively. In most cases, the influence of the pollutant variables ($PM_{10}$ and $PM_{2.5}$) was greater than that of the

meteorological parameters. However, at Daejeon, the most influential parameter on the $PM_{10}$ predictions was wind direction (45.67 %), while the contributions of other parameters were relatively small. The difference in the contributions is mainly due to the persistence of each variable. In other words, the variables with low dependence on the $PM_{10}$ and $PM_{2.5}$ predictions were those that change rapidly in the atmosphere, and thus their effects are scarcely incorporated into the trained model.

**4 Outlook and future works**

In this study, we established a deep RNN system for daily $PM_{10}$ and $PM_{2.5}$ predictions and evaluated the newly developed system's performance by comparing its $PM_{10}$ and $PM_{2.5}$ predictions with the observed and CMAQ-predicted levels. In the comparisons, the LSTM-based PM predictions were, in general, superior to the CMAQ-based PM predictions. In terms of IOA, the accuracies of the LSTM predictions were 1.01-1.72 times higher than those for the CMAQ-based predictions. Based on this, we concluded that the LSTM-based system could be applied to daily "operational" $PM_{10}$ and $PM_{2.5}$ forecasts.

The LSTM-based predictions at the observation sites can provide useful and complementary information for air quality





forecasters, synthesizing all the information available such as CTM air quality predictions, AI predictions, weather predictions, and satellite-derived information.

In the future, Korea's air quality forecasting system will be improved by continuous development of CTM-based prediction system including the uses of more advanced DA techniques, together with continuous sophistication of AI-based prediction

system. If the AI-based predictions at the observation sites are consistently better than the CTM-based predictions, the two elements will be more systematically combined within a prognostic mode, which will be our final research goal. In addition, a similar LSTM-based prediction system can also be applied to the daily forecasts of gas-phase air pollutants such as $NO_2$, $SO_2$, CO, and $O_3$. These works are also now in progress.

Although the current LSTM-based system can accurately predict $PM_{10}$ and $PM_{2.5}$, it also has some limitations. One, for better

prediction accuracy, we need more air quality data for model optimization. Because $PM_{2.5}$ has only been monitored in South Korea since 2015, there are too few observations to optimize the $PM_{2.5}$ predictions, which require continuous accumulation of $PM_{2.5}$ observations. In addition, the limited number of input variables is another obstacle to optimal model performance. The current LSTM-based $PM_{10}$ and $PM_{2.5}$ prediction system contains 10-12 input parameters. If more useful parameters such as mixing layer height (MLH) and barometric distribution are available, its performance would improve further (Hooyberghs

et al., 2005; Liu et al., 2007). Therefore, future efforts should be made with more $PM_{2.5}$ data and more input variables such as mixing layer heights entered into our system.

**Author contributions.** HSK and IP contributed equally to the paper. HSK and IP led the manuscript writing and contributed to the research design and system development. CHS supervised this study, contributed to the research design and manuscript writing, and served as the corresponding author. KL contributed to the CMAQ simulations. JWY, HKK, MJ, and
20 JL contributed to the optimization of the deep LSTM model.

**Acknowledgements.** This work was supported by the National Research Foundation of Korea (NRF) grant funded by the Korean Ministry of Science and ICT (MSIT) (NRF-2016R1C1B1012979) and by the National Strategic Project-Fine particle of the NRF funded by the MSIT, the Ministry of Environment (ME), and the Ministry of Health and Welfare (MOHW) (NRF-2017M3D8A1092022). We obtained NIER AIRKOREA and KMA AWS monitoring data from the official data
archives at https://www.airkorea.or.kr/ and https://data.kma.go.kr/, respectively.



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



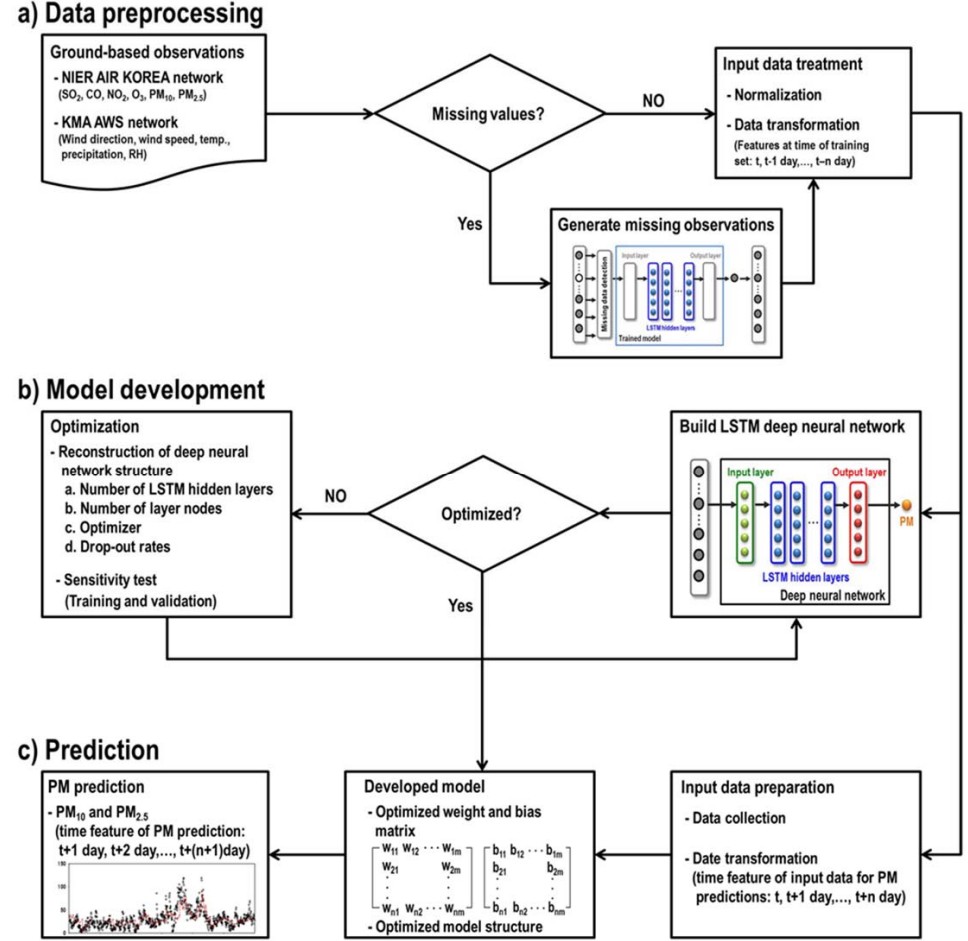

**Figure 1.** Schematic diagram of how the LSTM-based PM$_{10}$ and PM$_{2.5}$ prediction system was developed.



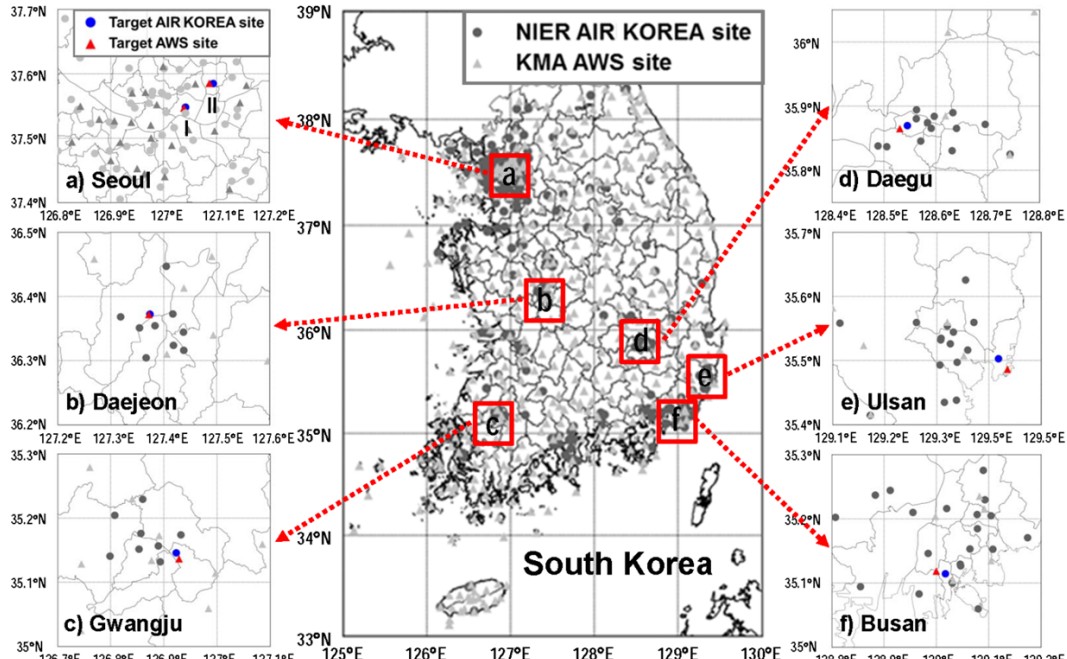

**Figure 2.** Locations of NIER AIR KOREA and KMA AWS sites in South Korea.





**Figure 3.** Training and validating the daily PM$_{10}$ prediction model: (a) Seoul-1; (b) Seoul-2; (c) Daejeon; (d) Gwangju; (e) Daegu; (f) Ulsan; (g) Busan. Black and red dots represent the training and validation results, respectively.

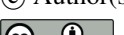


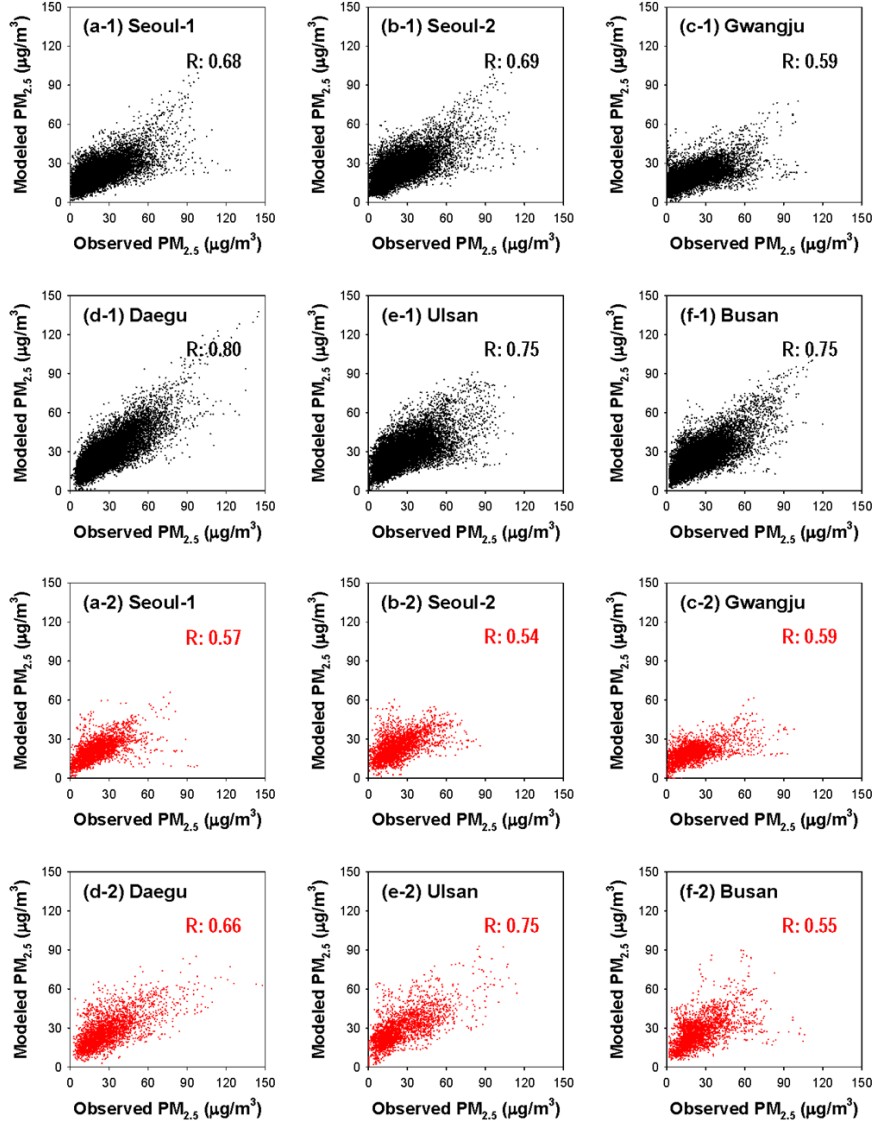

**Figure 4.** As Figure 3, except for PM$_{2.5}$.




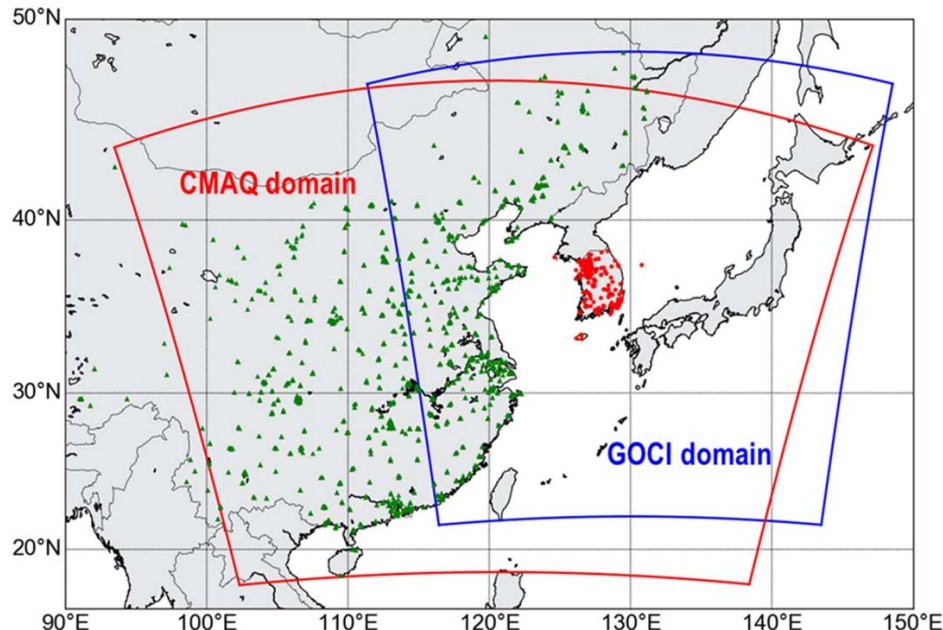

**Figure 5**. Domains of CMAQ model simulations (red line) and GOCI sensor (blue line). Green triangles and red dots represent the locations of ground-based monitoring sites in China and South Korea, respectively.



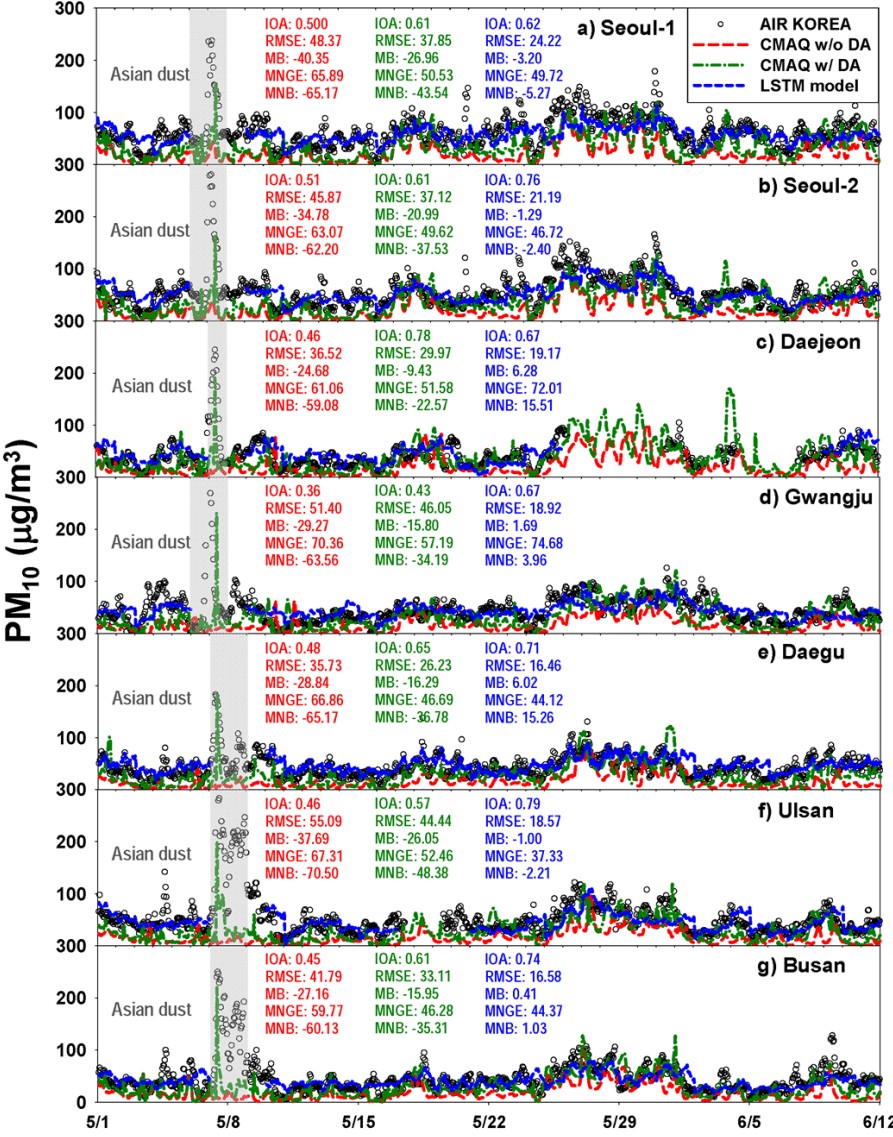

**Figure 6.** Comparisons between the CMAQ-calculated and LSTM-predicted and the observed PM$_{10}$. Black open circles show observed PM$_{10}$ at seven sites. Green- and red-dashed lines represent CMAQ-predicted PM$_{10}$ with and without data assimilation. Blue-dashed lines represent LSTM-predicted PM$_{10}$.



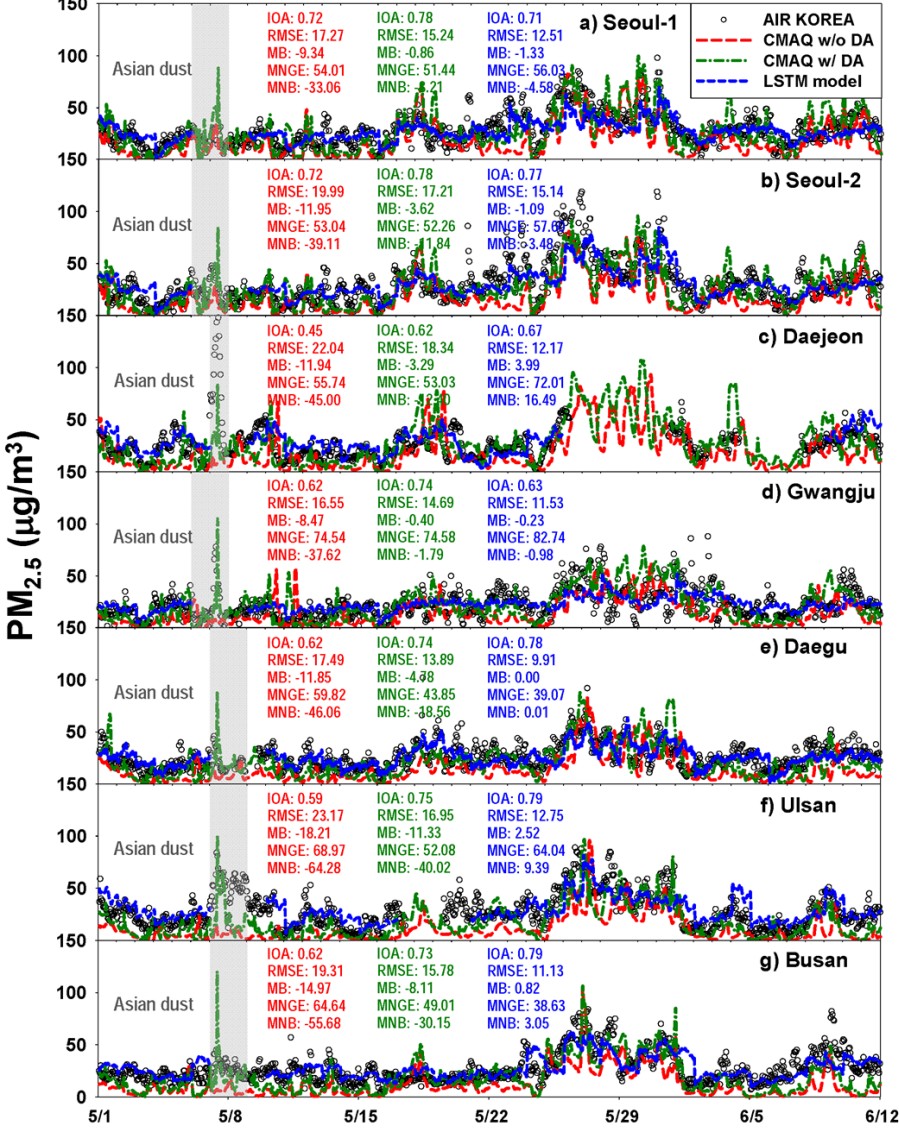

**Figure 7.** As Figure 6, except for PM$_{2.5}$.





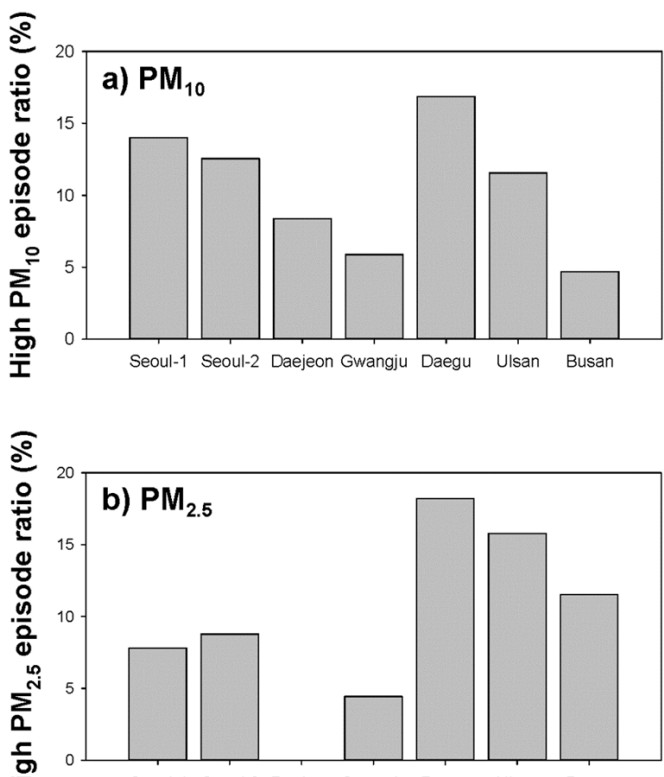

**Figure 8.** Percentage (%) of high particulate matter episodes in the training data set.



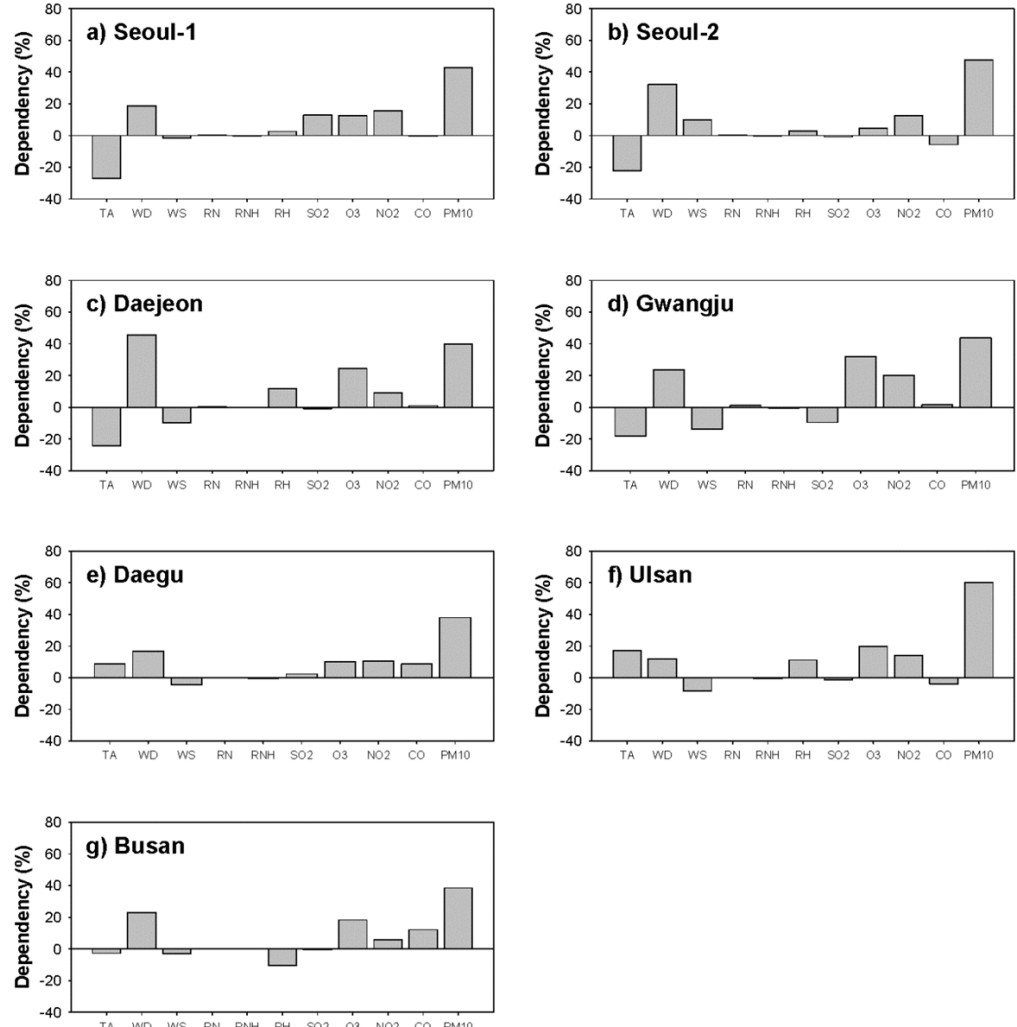

**Figure 9**. Dependencies of input variables on the daily PM$_{10}$ predictions. TA, WD, WS, RN, RNH, and RH represent temperature, wind direction, wind speed, daily cumulative precipitation, hourly precipitation, and relative humidity, respectively. SO$_2$, O$_3$, NO$_2$, CO, PM$_{10}$, and PM$_{2.5}$ refer to the previous day's levels of these atmospheric pollutants.



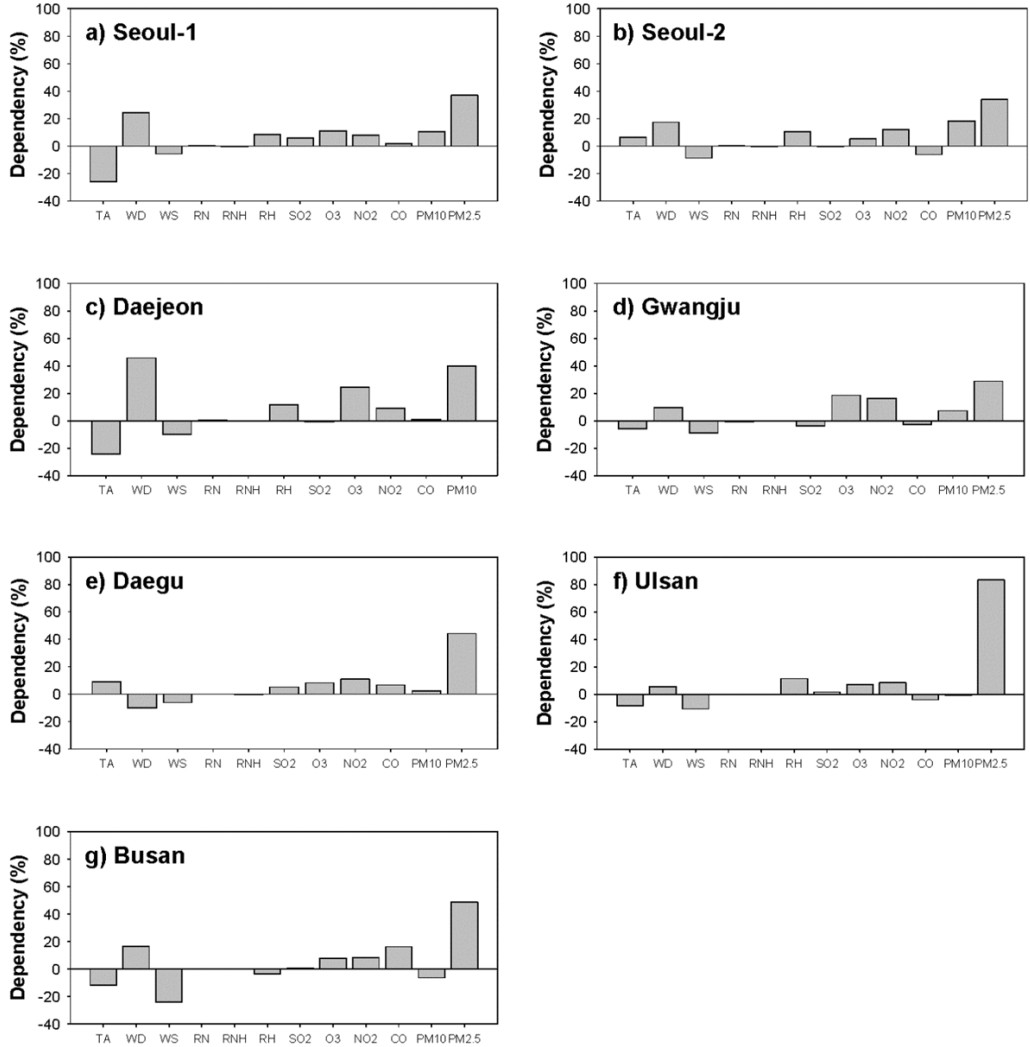

**Figure 10.** As Figure 9, except for PM$_{2.5}$.



Table 1. Summary of model training and validation results[1)]

| Species | Site | Training | | | Validation | | |
|---|---|---|---|---|---|---|---|
| | | R | MSE | RMSE | R | MSE | RMSE |
| $PM_{10}$ | Seoul-1 | 0.66 | 379.63 | 19.48 | 0.71 | 589.83 | 24.29 |
| | Seoul-2 | 0.72 | 347.62 | 18.65 | 0.59 | 551.71 | 23.49 |
| | Daejeon | 0.74 | 303.48 | 17.42 | 0.61 | 471.32 | 21.71 |
| | Gwangju | 0.61 | 326.98 | 18.08 | 0.55 | 362.88 | 19.05 |
| | Daegu | 0.81 | 259.39 | 16.11 | 0.63 | 378.18 | 19.45 |
| | Ulsan | 0.72 | 318.41 | 17.84 | 0.57 | 382.37 | 19.55 |
| | Busan | 0.71 | 230.59 | 15.19 | 0.55 | 394.44 | 19.86 |
| $PM_{2.5}$ | Seoul-1 | 0.68 | 106.73 | 10.33 | 0.57 | 118.24 | 10.87 |
| | Seoul-2 | 0.69 | 121.07 | 11.00 | 0.54 | 131.36 | 11.46 |
| | Daejeon | - | - | - | - | - | - |
| | Gwangju | 0.59 | 112.61 | 10.61 | 0.59 | 151.99 | 12.33 |
| | Daegu | 0.80 | 92.32 | 9.61 | 0.66 | 179.67 | 13.40 |
| | Ulsan | 0.75 | 145.16 | 12.05 | 0.75 | 165.26 | 12.86 |
| | Busan | 0.75 | 107.63 | 10.37 | 0.55 | 170.53 | 13.06 |

[1)] The units for MSE and RMSE are $\mu g/m^3$.





Table 2. Statistical analysis with modeled and observed $PM_{10}$ and $PM_{2.5}$[1]

| Station | Species | Model | Statistical parameter | | | | |
|---|---|---|---|---|---|---|---|
| | | | IOA | RMSE | MB | MNGE | MNB |
| Seoul - 1 | $PM_{10}$ | CMAQ w/o DA | 0.50 | 48.37 | -40.35 | 65.89 | -65.17 |
| | | CMAQ w/ DA | 0.61 | 37.85 | -26.96 | 50.53 | -43.54 |
| | | LSTM | 0.62 | 24.22 | -3.20 | 49.72 | -5.27 |
| | $PM_{2.5}$ | CMAQ w/o DA | 0.72 | 17.27 | -9.34 | 54.01 | -33.06 |
| | | CMAQ w/ DA | 0.78 | 15.24 | -0.86 | 51.44 | -3.21 |
| | | LSTM | 0.71 | 12.51 | -1.33 | 56.03 | -4.58 |
| Seoul - 2 | $PM_{10}$ | CMAQ w/o DA | 0.51 | 45.87 | -34.78 | 63.07 | -62.20 |
| | | CMAQ w/ DA | 0.61 | 37.12 | -20.99 | 49.62 | -37.53 |
| | | LSTM | 0.76 | 21.19 | -1.29 | 46.72 | -2.40 |
| | $PM_{2.5}$ | CMAQ w/o DA | 0.72 | 19.99 | -11.95 | 53.04 | -39.11 |
| | | CMAQ w/ DA | 0.78 | 17.21 | -3.62 | 52.26 | -11.84 |
| | | LSTM | 0.77 | 15.14 | -1.09 | 57.60 | -3.48 |
| Daejeon | $PM_{10}$ | CMAQ w/o DA | 0.46 | 36.52 | -24.68 | 61.06 | -59.08 |
| | | CMAQ w/ DA | 0.78 | 26.97 | -9.43 | 51.58 | -22.57 |
| | | LSTM | 0.67 | 19.17 | 6.28 | 72.01 | 15.51 |
| | $PM_{2.5}$ | CMAQ w/o DA | 0.45 | 22.04 | -11.94 | 55.74 | -45.00 |
| | | CMAQ w/ DA | 0.62 | 18.40 | -3.29 | 53.03 | -12.40 |
| | | LSTM | 0.67 | 12.17 | 3.99 | 72.01 | 16.49 |
| Gwangju | $PM_{10}$ | CMAQ w/o DA | 0.36 | 51.40 | -29.27 | 70.36 | -63.56 |
| | | CMAQ w/ DA | 0.43 | 46.05 | -15.80 | 57.19 | -34.19 |
| | | LSTM | 0.67 | 18.92 | 1.69 | 74.68 | 3.96 |
| | $PM_{2.5}$ | CMAQ w/o DA | 0.62 | 16.55 | -8.47 | 74.54 | -37.62 |
| | | CMAQ w/ DA | 0.74 | 14.69 | -0.40 | 74.58 | -1.79 |
| | | LSTM | 0.63 | 11.53 | -0.23 | 82.74 | -0.98 |
| Daegu | $PM_{10}$ | CMAQ w/o DA | 0.48 | 35.73 | -28.84 | 66.86 | -65.17 |
| | | CMAQ w/ DA | 0.65 | 26.23 | -16.29 | 46.69 | -36.78 |
| | | LSTM | 0.71 | 16.46 | 6.02 | 44.12 | 15.26 |
| | $PM_{2.5}$ | CMAQ w/o DA | 0.62 | 17.49 | -11.85 | 59.82 | -46.06 |
| | | CMAQ w/ DA | 0.74 | 13.89 | -4.78 | 43.85 | -18.56 |
| | | LSTM | 0.78 | 9.91 | 0.00 | 39.07 | 0.01 |
| Ulsan | $PM_{10}$ | CMAQ w/o DA | 0.46 | 55.09 | -37.69 | 67.31 | -70.50 |
| | | CMAQ w/ DA | 0.57 | 44.44 | -26.05 | 52.46 | -48.38 |
| | | LSTM | 0.79 | 18.57 | -1.00 | 37.33 | -2.21 |
| | $PM_{2.5}$ | CMAQ w/o DA | 0.59 | 23.17 | -18.21 | 68.97 | -64.28 |
| | | CMAQ w/ DA | 0.75 | 16.95 | -11.33 | 52.08 | -40.02 |
| | | LSTM | 0.79 | 12.75 | 2.52 | 64.04 | 9.39 |
| Busan | $PM_{10}$ | CMAQ w/o DA | 0.45 | 41.79 | -27.16 | 59.77 | -60.13 |
| | | CMAQ w/ DA | 0.61 | 33.11 | -15.95 | 46.28 | -35.31 |
| | | LSTM | 0.74 | 16.58 | 0.41 | 44.37 | 1.03 |
| | $PM_{2.5}$ | CMAQ w/o DA | 0.62 | 19.31 | -14.97 | 64.64 | -55.68 |
| | | CMAQ w/ DA | 0.73 | 15.78 | -8.11 | 49.01 | -30.15 |
| | | LSTM | 0.79 | 11.13 | 0.82 | 38.63 | 3.05 |

[1] The units for RMSE and MB are $\mu g/m^3$, and those for MNGE and MNB are in %.