# Peer review of "Development of daily PM10 and PM2.5 prediction system using a deep long short-term memory neural network model"

_Atmospheric Chemistry and Physics, 2019_

## Referee Comment (RC1) · Anonymous Referee #1 · 22 Apr 2019

**Review of 'Development of daily PM10 and PM2.5 prediction system using a deep long short-term memory neural network model by Kim et al.**

Kim et al. describe the development of a long short-term memory (LSTM) artificial recurrent neural network model trained to predict concentrations of PM10 and PM2.5 at seven Korean sites. They find that their LSTM model predicts PM concentrations with an accuracy better than (or at least comparable to) a chemical transport model.

The use of machine learning for air quality applications is a hot topic and the use of LSTM for the local prediction of PM2.5/PM10 is an interesting idea. As such, the manuscript offers a meaningful addition to the discussion on machine learning in atmospheric chemistry. The paper is well written and generally easy to follow. However, in its current form the manuscript lacks context and the quality of the LSTM is not entirely convincing. The following points need to be addressed in more detail before the paper can be recommended for publication:

1. The authors should better clarify what the intended use-case is for the machine learning model. In particular, there is some ambiguity in the word 'prediction' as it means different things in machine learning (where it means the 'guess' of the model) and atmospheric chemistry (where it refers to concentration estimates in the future): is the goal of the LSTM model to make a prediction of PM2.5/PM10 concentrations 24 hours from now - based on current conditions? If so, I assume the inputs/outputs have been prepared in such a way that they incorporate this 24-hour time lag? Or is the LSTM designed to make an optimal prediction of the concentration at a given time based on current conditions? In this case, it is not fully clear what the use-case for such a model would be.

In general, the mapping between input features and the predictor variable needs more explanation. For example, based on Figures 9 and 10 one would conclude that the variables needed to make a model prediction are the current meteorological conditions as well as the previous day pollutant concentrations? If this is the case, was there a rationale for this choice? Also, PM2.5 concentration is not used as an input for the PM10 prediction model (Figure 9), but seems to be used for the prediction of PM2.5 (Figure 10)?

2. The motivation to choose LSTM over another architecture should be discussed in more detail. Was LSTM selected because urban PM concentrations are expected to be dominated by local processes (e.g., emissions) and thus have a local, time-persistent signal? This would be a reasonable argument, but possibly also limits the usefulness of this approach to (urban) areas where PM concentrations are primarily determined by local processes? Based on the current version of the manuscript, it is not obvious why a simpler architecture (e.g., XGBoost) wouldn't yield a comparable (or even better) result.

3. The authors should clarify whether they trained just one LSTM model (for all 7 locations combined) or an individual LSTM for each station. If the former, can the LSTM model then also be used for PM predictions for a different city? This would be a powerful argument for this methodology and worthwhile testing.

4. Did the authors consider to use the logarithmic of $NO_2$ and $SO_2$ before normalizing the inputs? These species are often log-normally distributed and applying the regular normalization function to them (Eq. 1) might not be optimal. The generated missing values for both $NO_2$ and $SO_2$ are much worse than the predictions for the other four species (Figure S2), which might be further indication that these two species are not treated optimally. At the very least, a justification for using non-logarithmic concentration values for $NO_2$ and $SO_2$ should be provided.

5. The paragraph on model overfitting is confusing (page 6, line 14ff.): an overfitted model will produce better skill scores against the training data vs. the validation data since it has learned to fit well to the training data, but the model doesn't generalize. The results shown in Table 1 are not particularly encouraging in that regard and need more explanation.
It would also be helpful to provide more information on the network architecture, in particular the number of hidden nodes. Given that the number of input features is relatively small (11 variables per station per hour) and the training period only covers 2.3 years, it seems plausible that a complex model with too many modes will (a) overfit or (b) not converge to a (local) minimum in time because the training sample is too small. With regards to the latter, it would be instructive to show the MSE as a function of training cycles.

6. Another issue that should be addressed in the context of overfitting is the correlation of input variables: I assume some of the input features are highly correlated (e.g. $NO_2$ and $SO_2$, PM2.5 and PM10, temperature and $O_3$, etc.). While this is not a problem for the LSTM, per se, it lowers the amount of (independent) information contained in the training data and will likely slow convergence of the LSTM model as the model 'wastes time' learning these correlations first.
In that regard it is surprising to see that, for a number of stations, the PM2.5 prediction strongly depends on the previous day PM2.5 concentration but shows little dependency (or even a negative dependency) on PM10 concentration (Figure 10). Is this an expected result?

7. The CTM used in this study was run at 15x15 $km^2$ horizontal resolution, which can make it challenging to compare its output against ground-based observations due to representation error. This is particularly true for urban sites that might be heavily influenced by local, small-scale emission sources that are difficult to capture at this model resolution. As such, the comparison between CTM vs. LSTM predictions is somewhat unfair as it seems likely that a CTM with a local bias correction applied to it would perform significantly better. While this might be difficult to quantify, it should at least be addressed in the revised version of the manuscript.

**Minor comments:**

- Page 4, line 12: it would be helpful to provide the number of missing values (in %) for the pollutant concentrations.

- Page 6, line 17: I assume the authors mean 'overtuned', not 'overturned'…

- Page 21/22: the authors should explain why the LSTM predictions are missing for Daejeon from approximately 5/27 to 6/7.

- Appendix, equation S4: Isn't the sigmoid function defined as: $\sigma(x) = 1 / ( 1 + e^{-x} )$ ?

---

## Referee Comment (RC2) · Anonymous Referee #2 · 10 Jun 2019

Kim et al. developed LSTM-based PM10/PM2.5 prediction model and showed better performance than CTM. It is surprising that PM2.5/PM10 is more accurately predicted by a machine learning-based model using a few variables than by a complicated 3-dimensional chemical transport model incorporating emission, chemical production/loss, transport and diffusion, wet/dry deposition processes. Especially, this paper is meaningful that machine learning is applied to PM10/PM2.5 forecasting in Korea where the terrain is very complicated, and PM10/PM2.5 comes from diverse sources and is  frequently influenced from continental pollution. I have only a few points for better clarification.

1. Authors selected variables for machine learning using their knowledges and experiences. However, the square of the pearson correlation coefficient ($R^2$) in Fig 3. and 4 looks not greater than 0.5. meaning that the input variables have only 50% of explanatory power. Can this not limit the performance of machine learning based model?

2. In major cities in Korea, NO2 and CO are likely to be correlated due to share the common emission source. Does the dependency between input variables worsen LSTM performance or have little effect on it?

3. The high pollution events of PM10/PM2.5 in Korea are usually caused by long-range transport(LRT) and atmospheric congestion(AC). In most cases both LRT and AC play a role sequentially in polluted days. However, LSTM showed poor prediction at LRT case of May 25 to 28, 2016. Did authors consider any other model or any combination of LSTM and CNN(or DNN) in order to capture both LRT and AC?

4. Air quality forecasting is usually intended for high pollution events. Did authors consider to estimate the LSTM by categorical statistics such as critical success index(CSI), probability of detection(POD), false alarm ratio(FAR), and etc? If then, as high pollution events are not frequent, did authors consider the issue of data imbalance between normal and polluted days?

5. Several things such as data representation, activation function, weight initialization, pre-processing, hyper parameter are important for

determining machine learning model. I believe that authors performed a number of test to find the optimal method. Did authors not present for any reason all the information about them?

6. Correction of missing data is very important, especially, in machine learning algorithm. Authors developed the pre-trained deep LSTM model in order to generate missing data. As a result, the performance of the pre-trained deep LSTM model varies considerably with pollutant species. Does this affect the low dependance of SO2 and NO2 on PM10/PM2.5 prediction or not?

---

## Author Comment (AC1) · 7 Aug 2019

**Response to reviewer's comments:**

First of all, thank you for your valuable comments and suggestions. In the revised manuscript, we attempt to improve the manuscript based on your comments and suggestions. The added/modified parts are painted in a red color in the revised manuscript. Here, we would like to reply to some specific comments raised by you below:

**Author's comments to anonymous referee #1**

1. *"The authors should better clarify what the intended use-case is for the machine learning model. In particular, there is some ambiguity in the word 'prediction' as it means different things in machine learning (where it means the 'guess' of the model) and atmospheric chemistry (where it refers to concentration estimates in the future): is the goal of the LSTM model to make a prediction of $PM_{2.5}$/$PM_{10}$ concentrations 24 hours from now - based on current conditions? If so, I assume the inputs/outputs have been prepared in such a way that they incorporate this 24-hour time lag? Or is the LSTM designed to make an optimal prediction of the concentration at a given time based on current conditions? In this case, it is not fully clear what the use-case for such a model would be. In general, the mapping between input features and the predictor variable needs more explanation. For example, based on Figures 9 and 10 one would conclude that the variables needed to make a model prediction are the current meteorological conditions as well as the previous day pollutant concentrations? If this is the case, was there a rationale for this choice? Also, $PM_{2.5}$ concentration is not used as an input for the $PM_{10}$ prediction model (Figure 9), but seems to be used for the prediction of $PM_{2.5}$ (Figure 10)?"*

Reply) The aim of the LSTM system is daily prediction of $PM_{10}$ and $PM_{2.5}$ with one hour resolution. In this study, we used the meteorological conditions and atmospheric pollutant concentrations of the previous day as input parameters to predict the next 24 hour PM concentrations (i.e., there is a 24-hr time lag between the independent and dependent variables). Regarding this, we added a brief explanation about the mapping between the input and target values in the revised manuscript (please, refer to pp. 4:30-5:2).

Based on the reviewer's comment, we conducted a sensitivity test for the new input variable of $PM_{2.5}$ to the $PM_{10}$ predictions. The accuracy of $PM_{10}$ predictions with and without the added independent variable of $PM_{2.5}$ is summarized in Table R1. Even with the $PM_{2.5}$, no significant changes were found in the LSTM-based $PM_{10}$ predictions. In terms of IOA, most of the LSTM-based $PM_{10}$ predictions without $PM_{2.5}$ are correlated better with the observed $PM_{10}$ than did those with $PM_{2.5}$ except for the Seoul-2 and Daegu sites. Several AIR KOREA sites did not have monitored $PM_{2.5}$. In addition, this variable is not always available for the $PM_{10}$ predictions because of instrument malfunction such as at the Daejeon site. Therefore, the current combination of independent variables is more suitable for daily $PM_{10}$ prediction, we believe (please, also check out Table R1 attached in this reply).

2. "*The motivation to choose LSTM over another architecture should be discussed in more detail. Was LSTM selected because urban PM concentrations are expected to be dominated by local processes (e.g., emissions) and thus have a local, time-persistent signal? This would be a reasonable argument, but possibly also limits the usefulness of this approach to (urban) areas where PM concentrations are primarily determined by local processes? Based on the current version of the manuscript, it is not obvious why a simpler architecture (e.g., XGBoost) wouldn't yield a comparable (or even better) result.*"

Reply) We added more explanations for why we selected the deep LSTM neural network to develop the daily PM prediction system in South Korea (please, see pp. 5:6-12)

3. "*The authors should clarify whether they trained just one LSTM model (for all 7 locations combined) or an individual LSTM for each station. If the former, can the LSTM model then also be used for PM predictions for a different city? This would be a powerful argument for this methodology and worthwhile testing.*"

Reply) We developed individual PM prediction models for seven selected sites (please, see pp. 3:15-17).

4. "*Did the authors consider to use the logarithmic of NO2 and SO2 before normalizing the inputs? These species are often log-normally distributed and applying the regular normalization function to them (Eq. 1) might not be optimal. The generated missing values for both NO2 and SO2 are much worse than the predictions for the other four species (Figure S2), which might be further indication that these two species are not treated optimally. At the very least, a justification for using non-logarithmic concentration values for NO2 and SO2 should be provided.*"

Reply) To optimize data preprocessing, we conducted several sensitivity tests including converting the input variables into logarithmic values. The generation accuracy of missing $SO_2$ and $NO_2$ with and without logarithmic conversion (LC) of $SO_2$ and $NO_2$ is summarized in Table R2. As shown in Table R2, in general the generated missing values without LC showed better correlations with the observations than did those with LC.

   In addition, we evaluated the impact of LC on the PM predictions. The performance of LSTM-based PM predictions with and without the LC of $SO_2$ and $NO_2$ is summarized in Table R3. The PM predictions without LC were more accurate ($0.62 \leq IOA_{PM10} \leq 0.79$; $0.63 \leq IOA_{PM2.5} \leq 0.79$) than were those with LC ($0.60 \leq IOA_{PM10} \leq 0.74$; $0.63 \leq IOA_{PM2.5} \leq 0.77$).

5. "*The paragraph on model overfitting is confusing (page 6, line 14ff.): an overfitted model will produce better skill scores against the training data vs. the validation data since it has learned to fit well to the training data, but the model doesn't generalize. The results shown in Table 1 are not particularly encouraging in that regard and need more explanation. It would also be helpful to provide more information on the network architecture, in particular the number of hidden nodes.*

*Given that the number of input features is relatively small (11 variables per station per hour) and the training period only covers 2.3 years, it seems plausible that a complex model with too many modes will (a) overfit or (b) not converge to a (local) minimum in time because the training sample is too small. With regards to the latter, it would be instructive to show the MSE as a function of training cycles."*

Reply) We added more detailed explanations for model training in the revised manuscript (please, refer to pp. 6:29 – pp. 7:10).

6. *"Another issue that should be addressed in the context of overfitting is the correlation of input variables: I assume some of the input features are highly correlated (e.g. $NO_2$ and $SO_2$, $PM_{2.5}$ and $PM_{10}$, temperature and $O_3$, etc.). While this is not a problem for the LSTM, per se, it lowers the amount of (independent) information contained in the training data and will likely slow convergence of the LSTM model as the model 'wastes time' learning these correlations first. In that regard it is surprising to see that, for a number of stations, the $PM_{2.5}$ prediction strongly depends on the previous day $PM_{2.5}$ concentration but shows little dependency (or even a negative dependency) on $PM_{10}$ concentration (Figure 10). Is this an expected result?"*

Reply) We added more detailed discussions about the multicollinearity in the revised manuscript (please, see pp. 5:27-6:4).

We also agree with the reviewer's opinion that the number of available independent variables is very important for improving the performances of the LSTM-based PM prediction model. We organized the current data set with 11 to 12 independent variables that all could be collected from the ground-based observations (i.e., AIR KOREA and AWS networks). As mentioned in Sect. 4, if more useful independent variables are obtained from ground observations, we expect that the performances of the LSTM-based model could be improved further. This is also our research topic in the future.

The model training is a process to find the optimized weight and bias vectors to minimize the outcome of cost function (i.e., general scientific knowledge is not taken into account in the model training). In the deep learning, it is impossible to identify the causal relationships between independent and dependent variables (black box). In general, the prediction of the deep neural network model is mainly determined by input features used in the model training. Therefore, it is expected that the unusual correlations between $PM_{10}$ and $PM_{2.5}$ would be originated from the principle of model training and the characteristics of training data.

7. *"The CTM used in this study was run at 15x15 km2 horizontal resolution, which can make it challenging to compare its output against ground-based observations due to representation error. This is particularly true for urban sites that might be heavily influenced by local, smallscale emission sources that are difficult to capture at this model resolution. As such, the comparison between CTM vs. LSTM predictions is somewhat unfair as it seems likely that a CTM with a local bias correction applied to it would perform significantly better. While this might be difficult to quantify, it should at*

*least be addressed in the revised version of the manuscript."*

Reply) This is well-known sub-grid variability problem!! We added discussion into pp. 9:9-17. Please, take a look at this part!

Minor comments:

*"Page 4, line 12: it would be helpful to provide the number of missing values (in %) for the pollutant concentrations "*

Reply) The number of missing observations is summarized in Table S1.

*"Page 6, line 17: I assume the authors mean 'overtuned', not 'overturned'"*

Reply) We corrected it (please, see pp. 7:9).

*"Page 21/22: the authors should explain why the LSTM predictions are missing for Daejeon from approximately 5/27 to 6/7.'"*

Reply) We added an explanation of why there were no $PM_{10}$ predictions on those days (please, see pp. 23).

*"Appendix, equation S4: Isn't the sigmoid function defined as: s(x) = 1 / ( 1 + e-x ) ?'"*

Reply) In this study, we used the hard sigmoid function to activate the LSTM gate. We corrected the description (please, check out pp. S2:13-14).

Table R1. Accuracy of the LSTM-based $PM_{10}$ predictions with and without the input variable of $PM_{2.5}$ [1]

| Site | Statistical parameter | Without $PM_{2.5}$ | With $PM_{2.5}$ |
|---|---|---|---|
| Seoul-1 | IOA | 0.62 | 0.61 |
| | RMSE | 24.22 | 23.81 |
| | MB | -3.2 | -2.81 |
| | MNGE | 49.72 | 49.50 |
| | MNB | -5.27 | -4.64 |
| Seoul-2 | IOA | 0.76 | 0.79 |
| | RMSE | 21.19 | 21.29 |
| | MB | -1.29 | 0.02 |
| | MNGE | 46.72 | 47.53 |
| | MNB | -2.4 | 0.04 |
| Daejeon | IOA | 0.67 | - |
| | RMSE | 19.17 | - |
| | MB | 6.28 | - |
| | MNGE | 72.01 | - |
| | MNB | 15.51 | - |
| Gwangju | IOA | 0.67 | 0.66 |
| | RMSE | 18.92 | 18.10 |
| | MB | 1.69 | -1.88 |
| | MNGE | 74.68 | 67.85 |
| | MNB | 3.96 | -4.39 |
| Daegu | IOA | 0.71 | 0.72 |
| | RMSE | 16.46 | 15.71 |
| | MB | 6.02 | 4.82 |
| | MNGE | 44.12 | 41.81 |
| | MNB | 15.26 | 11.45 |
| Ulsan | IOA | 0.79 | 0.76 |
| | RMSE | 18.57 | 17.25 |
| | MB | -1 | 0.08 |
| | MNGE | 37.33 | 34.05 |
| | MNB | -2.21 | 0.18 |
| Busan | IOA | 0.74 | 0.71 |
| | RMSE | 16.58 | 17.39 |
| | MB | 0.41 | 1.48 |
| | MNGE | 44.37 | 46.33 |
| | MNB | 1.03 | 3.69 |

[1] The units for RMSE and MB are $\mu g/m^3$, and those for MNGE and MNB are in %.

Table R2. Accuracy comparison of missing $SO_2$ and $NO_2$ generations using log and non-log scale $SO_2$ and $NO_2$[1]

| Site | | non-log scale | | | | | log scale | | | |
|---|---|---|---|---|---|---|---|---|---|---|
| | Statistics | Tr. $SO_2$ | Val. $SO_2$ | Tr. $NO_2$ | Val. $NO_2$ | Statistics | Tr. $SO_2$ | Val. $SO_2$ | Tr. $NO_2$ | Val. $NO_2$ |
| Seoul-1 | IOA | 0.74 | 0.46 | 0.92 | 0.88 | IOA | 0.74 | 0.47 | 0.90 | 0.88 |
| | RMSE | 1.83 | 1.93 | 9.48 | 9.87 | RMSE | 1.83 | 2.25 | 10.33 | 9.50 |
| | MB | 0.13 | -0.75 | 0.54 | -0.39 | MB | -0.46 | -1.37 | -1.46 | -1.89 |
| | MNGE | 43.75 | 24.62 | 26.55 | 26.77 | MNGE | 35.62 | 27.73 | 26.13 | 25.35 |
| | MNB | 2.72 | -17.00 | 1.49 | -1.26 | MNB | -9.64 | -26.56 | -4.07 | -5.96 |
| Seoul-2 | IOA | 0.84 | 0.56 | 0.91 | 0.88 | IOA | 0.74 | 0.51 | 0.90 | 0.88 |
| | RMSE | 1.13 | 1.32 | 8.15 | 9.69 | RMSE | 1.30 | 1.30 | 8.24 | 9.96 |
| | MB | -0.08 | 0.21 | 1.29 | 0.86 | MB | -0.44 | -0.26 | -1.33 | -2.51 |
| | MNGE | 14.52 | 18.31 | 21.05 | 27.30 | MNGE | 14.59 | 16.32 | 17.56 | 24.17 |
| | MNB | -1.36 | 3.50 | 3.19 | 2.41 | MNB | -7.72 | -4.38 | -7.24 | -7.06 |
| Daejeon | IOA | 0.93 | 0.84 | 0.84 | 0.77 | IOA | 0.92 | 0.81 | 0.85 | 0.76 |
| | RMSE | 0.79 | 0.86 | 6.03 | 6.09 | RMSE | 0.84 | 0.91 | 6.11 | 6.24 |
| | MB | -0.01 | -0.27 | 0.06 | -0.79 | MB | -0.14 | -0.32 | -0.66 | -1.20 |
| | MNGE | 27.08 | 21.93 | 54.69 | 45.76 | MNGE | 25.02 | 23.36 | 45.08 | 41.44 |
| | MNB | -0.19 | -9.65 | 0.45 | -7.03 | MNB | -5.46 | -11.52 | -5.43 | -10.79 |
| Gwangju | IOA | 0.87 | 0.67 | 0.79 | 0.79 | IOA | 0.75 | 0.59 | 0.77 | 0.79 |
| | RMSE | 1.06 | 1.11 | 9.07 | 9.38 | RMSE | 1.32 | 1.21 | 9.66 | 9.28 |
| | MB | -0.06 | -0.42 | -0.28 | 1.36 | MB | -0.40 | -0.65 | -0.56 | 0.49 |
| | MNGE | 24.21 | 23.74 | 37.55 | 49.96 | MNGE | 25.36 | 23.47 | 34.72 | 46.32 |
| | MNB | -1.64 | -13.83 | -7.46 | 6.41 | MNB | -11.81 | -21.24 | -13.75 | 2.33 |
| Daegu | IOA | 0.81 | 0.68 | 0.88 | 0.89 | IOA | 0.68 | 0.59 | 0.87 | 0.85 |
| | RMSE | 2.38 | 2.46 | 8.83 | 11.30 | RMSE | 2.72 | 2.55 | 9.08 | 12.25 |
| | MB | 0.08 | 0.01 | 0.66 | -5.08 | MB | -0.94 | -1.08 | -0.30 | -6.60 |
| | MNGE | 65.92 | 54.14 | 38.95 | 28.68 | MNGE | 48.15 | 43.04 | 37.11 | 29.65 |
| | MNB | 2.15 | 0.23 | 2.82 | -15.71 | MNB | -27.10 | -28.78 | -1.30 | -20.42 |
| Ulsan | IOA | 0.85 | 0.70 | 0.89 | 0.88 | IOA | 0.77 | 0.76 | 0.90 | 0.87 |
| | RMSE | 5.72 | 5.92 | 7.64 | 7.99 | RMSE | 6.46 | 4.61 | 7.37 | 8.13 |
| | MB | -0.22 | 1.65 | 0.77 | 2.51 | MB | -1.06 | 0.64 | -1.21 | 0.63 |
| | MNGE | 44.70 | 52.01 | 41.18 | 34.88 | MNGE | 36.62 | 41.42 | 29.06 | 31.05 |
| | MNB | -2.82 | 26.50 | 3.98 | 12.53 | MNB | -13.51 | 10.21 | -6.26 | 3.16 |
| Busan | IOA | 0.65 | 0.63 | 0.75 | 0.67 | IOA | 0.65 | 0.63 | 0.68 | 0.60 |
| | RMSE | 2.48 | 2.56 | 9.34 | 11.85 | RMSE | 2.51 | 2.38 | 10.04 | 13.08 |
| | MB | -0.23 | 0.58 | 0.13 | -2.78 | MB | -0.69 | 0.11 | -2.44 | -5.84 |
| | MNGE | 26.36 | 36.95 | 46.96 | 40.16 | MNGE | 23.07 | 30.56 | 41.69 | 37.52 |
| | MNB | -3.36 | 9.26 | 0.61 | -10.56 | MNB | -10.19 | 1.84 | -11.40 | -22.21 |

[1] Tr. and Val. represent training and validation; the units for RMSE and MB are $\mu g/m^3$, and those for MNGE and MNB are in %.

Table R3. Accuracy comparison of PM predictions using log and non-log scale $NO_2$ and $SO_2$[1]

| Station | Statistical parameter | non-log scale | | log scale | |
|---|---|---|---|---|---|
| | | $PM_{10}$ | $PM_{2.5}$ | $PM_{10}$ | $PM_{2.5}$ |
| Seoul-1 | IOA | 0.62 | 0.71 | 0.60 | 0.69 |
| | RMSE | 24.22 | 12.51 | 24.66 | 13.03 |
| | MB | -3.2 | -1.33 | 1.06 | 0.08 |
| | MNGE | 49.72 | 56.03 | 54.99 | 59.06 |
| | MNB | -5.27 | -4.58 | 1.74 | 0.27 |
| Seoul-2 | IOA | 0.76 | 0.77 | 0.72 | 0.76 |
| | RMSE | 21.19 | 15.14 | 22.25 | 15.92 |
| | MB | -1.29 | -1.09 | -3.90 | -3.56 |
| | MNGE | 46.72 | 57.6 | 46.00 | 53.65 |
| | MNB | -2.4 | -3.48 | -7.28 | -11.39 |
| Daejeon | IOA | 0.67 | - | 0.66 | |
| | RMSE | 19.17 | - | 17.57 | |
| | MB | 6.28 | - | 6.16 | |
| | MNGE | 72.01 | - | 52.56 | |
| | MNB | 15.51 | - | 20.84 | |
| Gwangju | IOA | 0.67 | 0.63 | 0.67 | 0.63 |
| | RMSE | 18.92 | 11.53 | 19.02 | 11.65 |
| | MB | 1.69 | -0.23 | -3.55 | -1.49 |
| | MNGE | 74.68 | 82.74 | 63.13 | 78.93 |
| | MNB | 3.96 | -0.98 | -8.31 | -6.52 |
| Daegu | IOA | 0.71 | 0.78 | 0.72 | 0.73 |
| | RMSE | 16.46 | 9.91 | 17.50 | 10.27 |
| | MB | 6.02 | 0.00 | 8.17 | -0.17 |
| | MNGE | 44.12 | 39.07 | 45.98 | 39.56 |
| | MNB | 15.26 | 0.01 | 19.40 | -0.66 |
| Ulsan | IOA | 0.79 | 0.79 | 0.74 | 0.77 |
| | RMSE | 18.57 | 12.75 | 16.93 | 12.35 |
| | MB | -1.00 | 2.52 | -0.60 | -1.41 |
| | MNGE | 37.33 | 64.04 | 32.63 | 50.31 |
| | MNB | -2.21 | 9.39 | -1.47 | -5.46 |
| Busan | IOA | 0.74 | 0.79 | 0.72 | 0.74 |
| | RMSE | 16.58 | 11.13 | 17.22 | 11.71 |
| | MB | 0.41 | 0.82 | 1.81 | -2.46 |
| | MNGE | 44.37 | 38.63 | 46.13 | 36.08 |
| | MNB | 1.03 | 3.05 | 4.50 | -9.12 |

[1] The units for RMSE and MB are μg/m$^3$, and those for MNGE and MNB are in %

---

## Author Comment (AC2) · 7 Aug 2019

**Response to reviewer's comments:**

First of all, thank you for your valuable comments and suggestions. In the revised manuscript, we attempt to improve the manuscript based on your comments and suggestions. The added/modified parts are painted in a red color in the revised manuscript. Here, we would like to reply to some specific comments raised by you below:

**Author's comments to anonymous referee #2**

1. *"Authors selected variables for machine learning using their knowledges and experiences. However, the square of the pearson correlation coefficient (R2) in Fig 3. and 4 looks not greater than 0.5. meaning that the input variables have only 50% of explanatory power. Can this not limit the performance of machine learning based model?"*

Reply) The predication accuracy of deep neural network models is, in general, known to be very high. The performances of these models are mainly determined by input data used in the model training. We organized the current data set with 11 to 12 independent variables that were all information that could be collected from the ground-based observations (i.e., AIR KOREA and KMA AWS networks). This indicates that we used almost all chemical and meteorological variables available from the observations. In the model training, we used the observations for a period of 2.3 years because there was limited data availability. We expect that the performance of the LSTM-based PM prediction model would improve if more independent variables were obtained from ground observations and longer time-series observations were utilized in the model optimization in the future.

2. *"In major cities in Korea, $NO_2$ and CO are likely to be correlated due to share the common emission source. Does the dependency between input variables worsen LSTM performance or have little effect on it?."*

Reply) We added more detailed discussions about the multicollinearity issue in the revised manuscript (please, see pp. 5:27-6:4).

3. *"The high pollution events of $PM_{10}$/$PM_{2.5}$ in Korea are usually caused by long-range transport(LRT) and atmospheric congestion(AC). In most cases both LRT and AC play a role sequentially in polluted days. However, LSTM showed poor prediction at LRT case of May 25 to 28, 2016. Did authors consider any other model or any combination of LSTM and CNN(or DNN) in order to capture both LRT and AC."*

Reply) We may be able to improve the performances of the LSTM-based PM prediction model by combining different types or methods of neural network model that can predict high pollution events more accurately. To develop these models (or methods), it is essential to

identify high PM episode events and collect more amounts of variables, but these preliminary studies require considerable time. One example is the balancing the data for better predictions of high PM events. This issue is discussed in reply 4. However, this was not working very well. We think this issue will be able to be covered by future work!!

4. *"Air quality forecasting is usually intended for high pollution events. Did authors consider to estimate the LSTM by categorical statistics such as critical success index(CSI), probability of detection(POD), false alarm ratio(FAR), and etc? If then, as high pollution events are not frequent, did authors consider the issue of data imbalance between normal and polluted days?."*
Reply) We added a more detailed discussion on data imbalance in the revised manuscript (please, refer to pp. 11:3-18).

5. *"Several things such as data representation, activation function, weight initialization, pre-processing, hyper parameter are important for determining machine learning model. I believe that authors performed a number of test to find the optimal method. Did authors not present for any reason all the information about them?"*
Reply) We carried out several pre-tests to find out the optimized structure of the deep LSTM model. Recent deep learning studies have not provided detailed information about determining model structure because such descriptions must be extensive. In addition, the structure of deep neural network should change according to the configuration of independent and dependent variables. Therefore, we did not describe these contents in the manuscript.

The results of important sensitivity tests to determine the structure of $PM_{2.5}$ prediction model for Seoul-1 site are presented in Fig. R1. As shown in Fig. R1, the validation cost of the LSTM model training was the lowest when there were 100 hidden nodes (i.e., hidden neurons) and 5 hidden layers. In addition, the deep LSTM model showed optimal performances, when ReLU was embedded as an activation function, which is similar to previous studies (Nair and Hinton, 2010). Recent studies rarely used the sigmoid function, because of the gradient vanishing problem. For weight initialization, we applied the Xavier algorithm. This initialization method finds the optimal initial weight vectors according to the structure of the deep neural network (Glorot and Bengio, 2010). Because we adopted ADAM as an optimizer, the learning rate, which determines adjustment rate of weight and bias, continuously changed to find the global minima (Kingma and Ba, 2015).

6. *"Correction of missing data is very important, especially, in machine learning algorithm. Authors developed the pre-trained deep LSTM model in order to generate missing data. As a result, the performance of the pre-trained deep LSTM model varies considerably with pollutant*

*species. Does this affect the low dependance of SO2 and NO2 on PM10/PM2.5 prediction or not??"*

Reply) As we described in Sec. 2.3, one of the main criteria in selecting the PM prediction sites was the number of missing observations. The percentage of missing observations at seven sites is summarized in Table S1. As shown in Table S1, the fractions of missing observations are relatively small. Therefore, the values generated by the pre-trained model are unlikely to affect the dependencies of atmospheric pollutants. In order to confirm this, we performed the LSTM model training without missing observations. The dependence of the independent variables on the PM predictions of the previous model (including the missing values generated by the pre-trained model) and the newly trained model (excluding the missing observations) is summarized in Table R1. As shown in Table R1, the dependencies of $SO_2$ and $NO_2$ were also low, although the missing observations were not considered in the model training. In addition, we compared the performances of both models to evaluate the effects of missing observations on the PM predictions. The prediction accuracy of the two models is summarized in Table R2. In general, the PM predictions by the previous model were superior to those by the newly trained model, except at Gwangju site. This is because the missing observations generated by the pre-trained model enabled us to train the LSTM model for more various atmospheric conditions.

**References**

Glorot, X., Bengio, Y.: Understanding the difficulty of training deep feedforward neural networks, in: Proceedings of the 13[th] International Conference on Artificial Intelligence and Statistics, Sardinia, Italy, 13-15 May, 249-256, 2010.

Kingma, D. and Ba, J.: Adam: A method for stochastic optimization, in: Proceedings of the 3rd International Conference on Learning Representations, San Diego, USA, 3-8 May, arXiv:1412.6980v9, 2015.

Nair, V. and Hinton, G. E.: Rectified linear units improve restricted Boltzmann machines, in: Proceedings of the 27th International Conference on Machine Learning, Haifa, Israel, 21-24 June, 432, 2010.

[Figure]

**Figure R1**. Results of sensitivity test to determine the structure of PM$_{2.5}$ prediction model for Seoul-1 site.

Table R1. Dependency comparison between the LSTM model with and without considerations of missing observations[1]

| Species | Station | Model | Input variable | | | | | | | | | | | |
|---|---|---|---|---|---|---|---|---|---|---|---|---|---|---|
| | | | TA | WD | WS | RN | RNH | RH | SO$_2$ | O$_3$ | NO$_2$ | CO | PM$_{10}$ | PM$_{2.5}$ |
| PM$_{10}$ | Seoul-1 | LSTM w/ MO | -26.93 | 18.78 | -1.44 | 0.20 | -0.27 | 2.65 | 12.92 | 12.47 | 15.64 | -0.35 | 42.98 | - |
| | | LSTM w/o MO | -19.25 | 16.98 | 4.30 | 0.48 | -0.13 | 2.81 | 12.58 | 11.84 | 16.35 | 2.22 | 36.75 | - |
| | Seoul-2 | LSTM w/ MO | -22.07 | 32.29 | 10.05 | 0.22 | -0.14 | 2.91 | -0.87 | 4.74 | 12.42 | -5.70 | 47.55 | - |
| | | LSTM w/o MO | -32.59 | 31.08 | 3.43 | 0.32 | -0.25 | 3.79 | 1.34 | 10.07 | 18.54 | -2.05 | 49.41 | - |
| | Daejeon | LSTM w/ MO | -24.17 | 45.67 | -9.97 | 0.23 | -0.25 | 11.72 | -0.90 | 24.37 | 9.23 | 0.98 | 39.97 | - |
| | | LSTM w/o MO | -16.74 | 22.07 | -15.13 | 0.79 | -0.33 | 18.57 | 1.28 | 26.08 | 5.65 | 2.83 | 43.93 | - |
| | Gwangju | LSTM w/ MO | -18.34 | 23.68 | -13.94 | 1.02 | -0.50 | - | -9.62 | 31.84 | 20.09 | 1.58 | 43.61 | - |
| | | LSTM w/o MO | -1.43 | 25.78 | -9.18 | 0.87 | -0.58 | - | -9.62 | 31.15 | 17.66 | 6.53 | 48.56 | - |
| | Daegu | LSTM w/ MO | 8.85 | 16.59 | -4.39 | -0.04 | -0.40 | - | 2.40 | 10.18 | 10.49 | 8.65 | 37.87 | - |
| | | LSTM w/o MO | 2.11 | 5.87 | -7.05 | -0.01 | -0.63 | - | 3.30 | 16.42 | 14.81 | 10.97 | 40.94 | - |
| | Ulsan | LSTM w/ MO | 17.19 | 11.93 | -8.32 | 0.20 | -0.51 | 11.13 | -1.39 | 19.78 | 14.16 | -3.99 | 60.12 | - |
| | | LSTM w/o MO | 16.19 | 14.95 | -6.33 | 0.03 | -0.50 | 5.10 | 0.31 | 23.72 | 11.47 | -6.26 | 47.84 | - |
| | Busan | LSTM w/ MO | -2.83 | 22.95 | -2.95 | -0.03 | -0.08 | -10.40 | -0.35 | 18.30 | 5.67 | 12.24 | 38.48 | - |
| | | LSTM w/o MO | 11.44 | 19.25 | -10.77 | -0.04 | -0.04 | -18.37 | 1.80 | 14.66 | 11.20 | 7.74 | 30.97 | - |
| PM$_{2.5}$ | Seoul-1 | LSTM w/ MO | -25.85 | 24.23 | -5.67 | 0.40 | -0.33 | 8.32 | 5.74 | 11.06 | 8.04 | 1.64 | 10.54 | 37.05 |
| | | LSTM w/o MO | -33.87 | 25.21 | -6.30 | 0.10 | -0.38 | 11.41 | -5.99 | 14.74 | 13.66 | 2.20 | 11.74 | 35.94 |
| | Seoul-2 | LSTM w/ MO | 6.46 | 17.38 | -8.70 | 0.14 | -0.16 | 10.28 | -0.36 | 5.24 | 12.21 | -6.17 | 18.29 | 33.89 |
| | | LSTM w/o MO | -21.21 | 20.68 | -8.84 | -0.10 | -0.18 | 10.33 | -3.38 | 3.75 | 10.72 | -6.73 | 16.32 | 28.33 |
| | Daejeon | LSTM w/ MO | -24.17 | 45.67 | -9.97 | 0.23 | -0.25 | 11.72 | -0.90 | 24.37 | 9.23 | 0.98 | 39.97 | - |
| | | LSTM w/o MO | -16.74 | 22.07 | -15.13 | 0.79 | -0.33 | 18.57 | 1.28 | 26.08 | 5.65 | 2.83 | 43.93 | - |
| | Gwangju | LSTM w/ MO | -5.86 | 9.68 | -8.93 | -0.49 | -0.49 | - | -3.92 | 18.55 | 16.29 | -2.82 | 7.27 | 28.80 |
| | | LSTM w/o MO | -1.31 | 13.15 | -3.54 | 1.25 | -0.58 | - | -13.49 | 26.10 | 18.18 | -5.90 | 9.69 | 31.37 |
| | Daegu | LSTM w/ MO | 9.05 | -10.16 | -6.13 | -0.04 | -0.38 | - | 5.20 | 8.28 | 10.90 | 6.74 | 2.14 | 44.11 |
| | | LSTM w/o MO | 11.93 | -0.96 | -10.68 | -0.06 | -0.52 | - | 1.52 | 7.96 | 2.22 | 11.78 | 9.80 | 32.66 |
| | Ulsan | LSTM w/ MO | -8.11 | 5.63 | -10.52 | 0.07 | -0.15 | 11.56 | 1.75 | 7.14 | 8.54 | -3.82 | -0.51 | 83.38 |
| | | LSTM w/o MO | 4.31 | 7.13 | -10.70 | 0.32 | -0.23 | 14.02 | -1.26 | 21.07 | 11.87 | -10.40 | -1.50 | 81.17 |
| | Busan | LSTM w/ MO | -11.75 | 16.47 | -24.01 | 0.19 | -0.06 | -3.59 | 0.96 | 7.83 | 8.29 | 16.23 | -6.36 | 48.77 |
| | | LSTM w/o MO | 17.44 | 8.70 | -25.10 | 0.06 | 0.12 | 1.80 | -2.13 | 7.65 | 6.35 | 8.99 | 1.49 | 58.50 |

[1] LSTM w/ MO and LSTM w/o MO represent the LSTM model with and without consideration of missing observations in the model training; TA, WD, WS, RN, RNH, and RH denote temperature, wind direction, wind speed, daily accumulative precipitation, hourly precipitation, and relative humidity of previous day; SO$_2$, O$_3$, NO$_2$, CO, PM$_{10}$, and PM$_2$ are the concentrations of the respective air pollutants on the previous day.

Table R2. Performance comparison between the LSTM model with and without considerations of missing observations[1]

| Station | Species | Model | Statistical parameter | | | | |
|---|---|---|---|---|---|---|---|
| | | | IOA | RMSE | MB | MNGE | MNB |
| Seoul - 1 | $PM_{10}$ | LSTM w/ MO | 0.62 | 24.22 | -3.20 | 49.72 | -5.27 |
| | | LSTM w/o MO | 0.57 | 25.35 | 0.91 | 57.16 | 1.49 |
| | $PM_{2.5}$ | LSTM w/ MO | 0.71 | 12.51 | -1.33 | 56.03 | -4.58 |
| | | LSTM w/o MO | 0.71 | 12.95 | -2.56 | 53.53 | -8.81 |
| Seoul - 2 | $PM_{10}$ | LSTM w/ MO | 0.76 | 21.19 | -1.29 | 46.72 | -2.40 |
| | | LSTM w/o MO | 0.69 | 23.53 | -4.90 | 48.35 | -9.14 |
| | $PM_{2.5}$ | LSTM w/ MO | 0.77 | 15.14 | -1.09 | 57.60 | -3.48 |
| | | LSTM w/o MO | 0.75 | 16.06 | -4.62 | 51.52 | -14.78 |
| Daejeon | $PM_{10}$ | LSTM w/ MO | 0.67 | 19.17 | 6.28 | 72.01 | 15.51 |
| | | LSTM w/o MO | 0.59 | 19.13 | -0.44 | 62.07 | -1.16 |
| | $PM_{2.5}$ | LSTM w/ MO | 0.67 | 12.17 | 3.99 | 72.01 | 16.49 |
| | | LSTM w/o MO | 0.59 | 12.15 | -0.28 | 62.07 | -1.16 |
| Gwangju | $PM_{10}$ | LSTM w/ MO | 0.67 | 18.92 | 1.69 | 74.68 | 3.96 |
| | | LSTM w/o MO | 0.72 | 18.41 | 0.74 | 66.60 | 1.72 |
| | $PM_{2.5}$ | LSTM w/ MO | 0.63 | 11.53 | -0.23 | 82.74 | -0.98 |
| | | LSTM w/o MO | 0.68 | 11.86 | 2.40 | 95.92 | 10.46 |
| Daegu | $PM_{10}$ | LSTM w/ MO | 0.71 | 16.46 | 6.02 | 44.12 | 15.26 |
| | | LSTM w/o MO | 0.71 | 16.51 | 5.34 | 43.30 | 12.67 |
| | $PM_{2.5}$ | LSTM w/ MO | 0.78 | 9.91 | 0.00 | 39.07 | 0.01 |
| | | LSTM w/o MO | 0.67 | 11.55 | 0.79 | 43.78 | 3.06 |
| Ulsan | $PM_{10}$ | LSTM w/ MO | 0.79 | 18.57 | -1.00 | 37.33 | -2.21 |
| | | LSTM w/o MO | 0.69 | 18.60 | -1.55 | 37.37 | -3.41 |
| | $PM_{2.5}$ | LSTM w/ MO | 0.79 | 12.75 | 2.52 | 64.04 | 9.39 |
| | | LSTM w/o MO | 0.72 | 12.95 | -0.14 | 57.43 | -0.21 |
| Busan | $PM_{10}$ | LSTM w/ MO | 0.74 | 16.58 | 0.41 | 44.37 | 1.03 |
| | | LSTM w/o MO | 0.68 | 17.62 | 1.89 | 48.21 | 4.69 |
| | $PM_{2.5}$ | LSTM w/ MO | 0.79 | 11.13 | 0.82 | 38.63 | 3.05 |
| | | LSTM w/o MO | 0.77 | 12.07 | 0.91 | 40.98 | 3.39 |

[1] LSTM w/ MO and LSTM w/o MO represent the trained LSTM model with and without missing observations; the units for RMSE and MB are $\mu g/m^3$, and those for MNGE and MNB are in %.

---

## Author Response (AR1)

**August 7, 2019**

Dr. David Topping

Atmospheric Chemistry and Physics

School of Earth and Environmental Sciences

University of Manchester, Centre for Atmospheric Science

Manchester M13 9PL, UK

*Attn*: Dr. David Topping (Co-Editor)

**Re: Manuscript (acp-2019-268)**

Dear Dr. David Topping:

Please find our revised manuscript titled "*Development of daily $PM_{10}$ and $PM_{2.5}$ prediction system using a deep long short-term memory neural network model*" and our responses to both reviewers' comments. First of all, we would like to express our thanks to both reviewers for their valuable suggestions and comments on our manuscript. Base on both reviewers' suggestion and comments, we further improved the manuscript. The modified/added parts are painted in a red color in the revised manuscript.

Again, thank you for your consideration of this manuscript, and we look forward to hearing from you regarding the final decision of this paper.

Sincerely,

**Chul H. Song (Corresponding Author)**

School of Earth Sciences and Environmental Engineering

Gwangju Institute of Science and Technology (GIST)

123 Cheomdangwagi-ro, Buk-gu, Gwangju 61005, Korea

---

## Author Response (AR2)

**August 28, 2019**

Dr. David Topping

Atmospheric Chemistry and Physics

School of Earth and Environmental Sciences

University of Manchester, Centre for Atmospheric Science

Manchester M13 9PL, UK

***Attn*:** Dr. David Topping (Co-Editor)

**Re: Manuscript (acp-2019-268)**

Dear Dr. David Topping:

Please find our revised manuscript titled "*Development of daily $PM_{10}$ and $PM_{2.5}$ prediction system using a deep long short-term memory neural network model*". Upon your request, we added code ability to the revised manuscript. The added parts are pained in a red color. Since this research has been supported by the Korean Ministry of Science and ICT (MSIT), the Ministry of Environment (ME), and the Ministry of Health and Welfare (MOHW), all the research results are owned by the South Korean government. Therefore, the release of the source should be approved by the government.

Again, thank you for your consideration and handling of this manuscript.

Sincerely,

**Chul H. Song (Corresponding Author)**

School of Earth Sciences and Environmental Engineering

Gwangju Institute of Science and Technology (GIST)

123 Cheomdangwagi-ro, Buk-gu, Gwangju 61005, Korea